# The neural crest is a source of mesenchymal stem cells with specialized hematopoietic stem cell niche function

**Joan Isern\*, Andrés García-García, Ana M Martín, Lorena Arranz, Daniel Martín-Pérez, Carlos Torroja, Fátima Sánchez-Cabo, Simón Méndez-Ferrer\***

Stem Cell Niche Pathophysiology Group, Centro Nacional de Investigaciones Cardiovasculares, Madrid, Spain

**Abstract** Mesenchymal stem cells (MSCs) and osteolineage cells contribute to the hematopoietic stem cell (HSC) niche in the bone marrow of long bones. However, their developmental relationships remain unclear. In this study, we demonstrate that different MSC populations in the developing marrow of long bones have distinct functions. Proliferative mesoderm-derived nestin$^-$ MSCs participate in fetal skeletogenesis and lose MSC activity soon after birth. In contrast, quiescent neural crest-derived nestin$^+$ cells preserve MSC activity, but do not generate fetal chondrocytes. Instead, they differentiate into HSC niche-forming MSCs, helping to establish the HSC niche by secreting Cxcl12. Perineural migration of these cells to the bone marrow requires the ErbB3 receptor. The neonatal Nestin-GFP$^+$ Pdgfrα$^-$ cell population also contains Schwann cell precursors, but does not comprise mature Schwann cells. Thus, in the developing bone marrow HSC niche-forming MSCs share a common origin with sympathetic peripheral neurons and glial cells, and ontogenically distinct MSCs have non-overlapping functions in endochondrogenesis and HSC niche formation.

## Introduction

Bone marrow stromal cells (BMSCs) are a heterogeneous population. The different mesenchymal cell types might either arise from a variety of resident progenitors or might ultimately be derived from a single population of rare MSCs (*Caplan, 1991*). In adult mammals, a multiple origin of skeletal MSCs is suggested by the distinct germ layer derivation of different bone structures, with craniofacial bones generated by the neuroectoderm, whereas the axial and appendicular bones are respectively derived from paraxial and lateral mesoderm. Mesoderm generates chondrocytes, which are progressively replaced by osteoblasts through the process of endochondral ossification (*Olsen et al., 2000*). MSCs share cell-surface markers and localization with pericytes, suggesting that some pericytes might be MSCs (*Crisan et al., 2008*). However, it remains unclear whether the bone marrow hosts ontogenically distinct MSCs in the same bones and whether they are endowed with specific functions.

In adult bone marrow, a variety of mesenchymal cells regulate HSCs (*Calvi et al., 2003*; *Zhang et al., 2003*; *Arai et al., 2004*; *Sacchetti et al., 2007*; *Chan et al., 2008*; *Naveiras et al., 2009*; *Mendez-Ferrer et al., 2010*; *Omatsu et al., 2010*; *Raaijmakers et al., 2010*). Nevertheless, the specialized functions and developmental origin of these cells are largely unknown. Adult HSCs are also regulated by certain other non-hematopoietic lineages, including endothelial cells (*Avecilla et al., 2004*; *Kiel et al., 2005*; *Ding et al., 2012*), sympathetic neurons and associated non-myelinating Schwann cells (*Katayama et al., 2006*; *Spiegel et al., 2007*; *Mendez-Ferrer et al., 2008*; *Yamazaki et al., 2011*), perivascular cells expressing the leptin receptor (*Ding et al., 2012*) and mesodermal derivatives (*Greenbaum et al., 2013*). However, the relationships and potential overlap among these populations remain unclear. It is also not known whether MSCs that form the HSC niche also generate other stromal cells or are a specialized population that arises earlier in embryogenesis and persists into adulthood.

**\*For correspondence:** joan.isern@cnic.es (JI); smendez@cnic.es (SM)

**Competing interests:** The authors declare that no competing interests exist.

**eLife digest** During the earliest phases of development, the embryo is formed by groups of stem cells that can develop into all the different types of tissue in the body—from bones to brain tissue. Later in life, small stockpiles of adult stem cells are found in various tissues and provide a reservoir of new cells available for replacing old or damaged cells. The most important source of blood stem cells is the bone marrow, which produces and stores cells that are capable of developing into blood and immune system cells. These processes are assisted by different bone marrow cells called stromal cells, which create a specialized local environment or 'niche'.

But are the stromal stem cells that form the skeleton the same ones that form this niche during development? Or do the various types of stromal stem cells develop from distinct groups of cells in the embryo? Furthermore, it is unclear which cells guide blood stem cells towards the forming bones.

Other types of cells, including some of the cells of the nervous system, can communicate with the stem cells in the adult marrow and influence their behavior. This led scientists to wonder whether the stem cells in the bone marrow niche and the cells that communicate with them developed from the same type of embryonic stem cell.

Isern et al. tracked down the developmental origins of different types of bone marrow stromal stem cells by examining the bone marrow from the long bones (for example, the bones in the leg) of unborn and infant mice. It turns out that not all stromal stem cells in the developing bone marrow are alike. In fact, one pool of stromal stem cells forms the skeleton and loses stem cell activity in the process. In contrast, a different population of stromal stem cells develops from the same group of embryonic cells that gives rise to the cells of the nervous system. The stromal stem cells in this second group function as a niche to recruit and store the incoming blood stem cells and retain their stem cell activity throughout life.

The findings of Isern et al. help to explain why the nervous system is able to communicate with stem cells in the adult marrow, and provide a model for understanding how stem cell niches in organs that contain nerve tissue are established.

In this study, we investigated the developmental origin and functions of MSCs in the primordial marrow of long bones. We show that, like peripheral neural and glial cells, HSC niche-forming MSCs in perinatal bone marrow arise from the trunk neural crest and make only a modest contribution to endochondrogenesis. Thus, whereas mesoderm-derived MSCs are mostly involved in endochondral ossification, neural crest-derived cells have a specialized function in establishing the HSC niche in the developing marrow of the same bones. These results provide compelling evidence for functional segregation of MSCs derived from different germ layers. The data also show that three HSC niche components—peripheral sympathetic neurons, Schwann cells, and MSCs—share a common origin.

## Results

### Fetal bone marrow nestin[+] cells are quiescent and distinct from osteochondral cells

In adult mouse bone marrow, stromal cells expressing the green fluorescent protein (GFP) driven by the regulatory elements of nestin promoter (Nes-GFP[+]) display features of both MSCs and HSC niche cells (*Mendez-Ferrer et al., 2010*). This finding prompted us to characterize Nes-GFP[+] cells during marrow development in limb bones. GFP[+] cells were already present in E16.5 bone marrow, associated preferentially with blood vessels infiltrating the cartilage scaffold (*Figure 1—figure supplement 1A–C*). At E18.5 Nes-GFP[+] cells were frequently associated with arterioles and sprouting endothelial cells within the osteochondral junction (*Figure 1A–C*). Fetal bone marrow Nes-GFP[+] cells were heterogeneous, composed of a majority of BMSCs but also including a small subset of CD31[+] putative endothelial cells that increased during the postnatal period (*Figure 1D,E* and *Figure 1—figure supplement 1D*). Compared with Nes-GFP[-] BMSCs, the Nes-GFP[+] cell population was enriched in endogenous *Nestin* mRNA expression (*Figure 1F*). Arterioles were associated with an intense fluorescence microscopy signal, due to the presence of several concentric GFP[+] cells, including an outer layer that expressed smooth

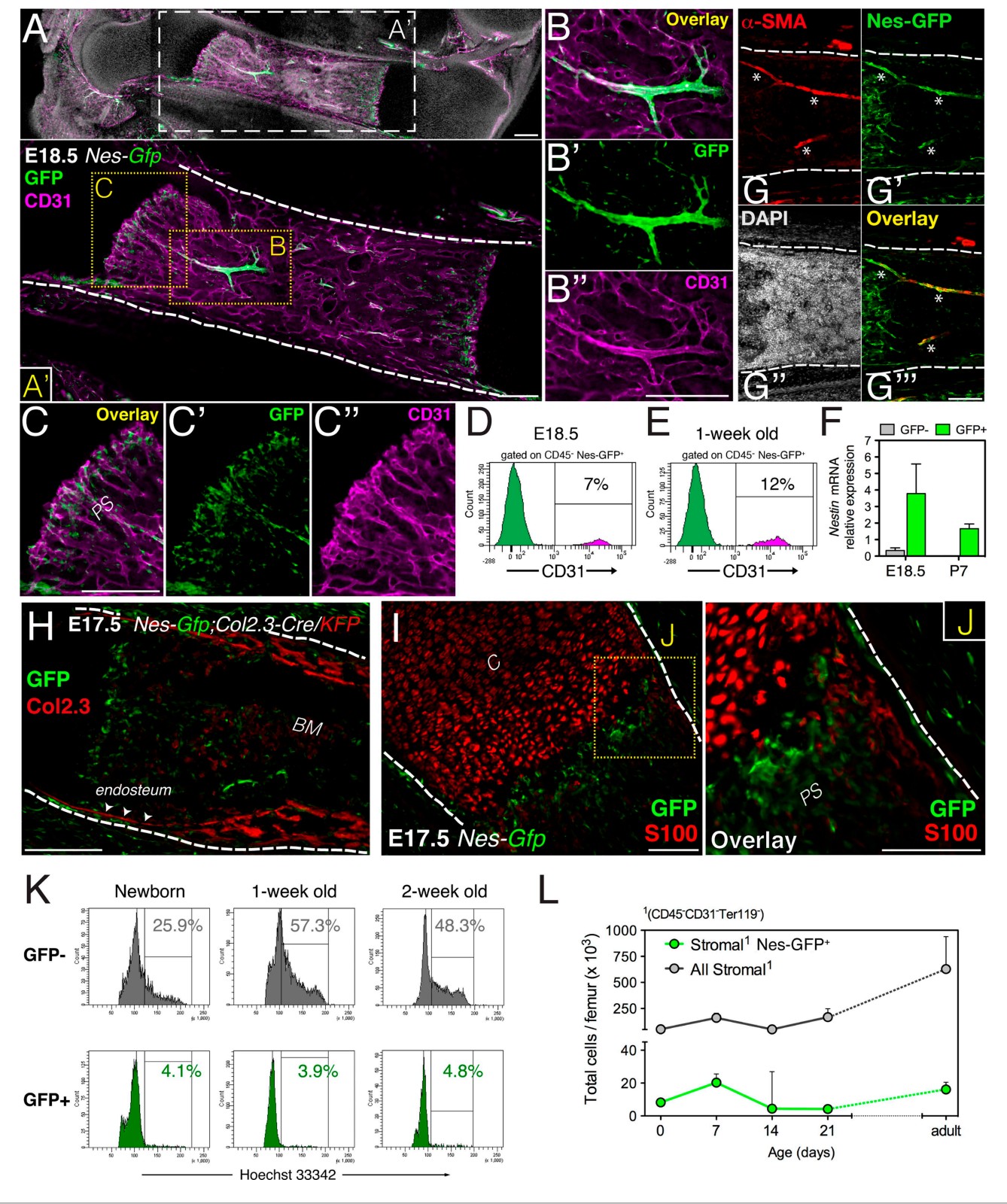

**Figure 1**. Fetal bone marrow nestin+ cells proliferate slowly and are distinct from osteochondral cells. (**A**–**C**) Nes-GFP+ cells in fetal bones undergoing endochondral ossification. Whole-mount confocal projection of E18.5 *Nes-Gfp* femoral bone marrow stained with CD31 (magenta) to mark endothelium. Note the perivascular distribution of GFP+ cells (green) in arterioles (**B**–**B''**) and small vessels invading the primary spongiosa (**C**–**C''**). (**D**–**E**) *Nes-Gfp* transgene is

*Figure 1. Continued on next page*

*Figure 1. Continued*

expressed by a subset of bone marrow endothelial cells. Flow cytometry histograms show the frequency of CD45⁻ Nes-GFP⁺ cells expressing CD31. (**F**) Endogenous *Nestin* mRNA expression measured by qPCR in stromal populations isolated from *Nes-Gfp* mice at the indicated stages (mean ± SD, *n* = 3–5). (**G**) *Nes-Gfp* bone marrow section stained with smooth muscle actin antibodies (αSma, red; asterisks) to reveal arterioles. (**H**) Limb section from an E17.5 *Nes-Gfp;Col2.3-Cre;KFP* embryo showing *Nes*-GFP⁺ (green) and osteoblasts identified with antibodies to Katushka (KFP) protein (red), driven by the 2.3-kb proximal fragment of the α1(I)-collagen promoter. Arrowheads, endosteal surface. (**I**) Metaphysis of E17.5 *Nes-Gfp* embryo showing S100⁺ chondrocytes (red). (**J**) Magnified view of boxed area in (**I**). (**K**) Representative cell cycle profiles of bone marrow stromal Nes-GFP⁺/⁻ cells at early postnatal stages. Frequencies of cells in $G_2$/S-M (%) are indicated. (**L**) Number of stromal Nes-GFP⁺/⁻ cells in postnatal bone marrow (mean ± SEM, *n* = 3–4). Scale bars: 200 µm (**A**, **A'**, **B''**, **C**, **H**), 100 µm (**G**, **I** and **J**); (**A'**, **G**–**J**) dashed line indicates bone contour. *BM*, bone marrow; *C*, cartilage; *PS*, primary spongiosa.

The following figure supplement is available for figure 1:

**Figure supplement 1**. Perivascular and endothelial Nes-GFP⁺ cells invade the incipient bone marrow associated with blood vessels.

muscle actin and an inner layer of endothelial cells (*Figure 1G* and *Figure 1—figure supplement 1E,F*). Fetal bone marrow Nes-GFP⁺ cells were distinct from S100-expressing chondrocytes and osteoblastic cells genetically labeled with the 2.3-kilobase proximal fragment of the α1(I)-collagen promoter (*Dacquin et al., 2002*) (*Figure 1H–J*). Contrasting the marked proliferation of Nes-GFP⁻ BMSCs in perinatal life, Nes-GFP⁺ cells remained mostly quiescent (*Figure 1K* and *Figure 1—figure supplement 1G*). As a result, whereas Nes-GFP⁻ BMSCs steadily expanded, Nes-GFP⁺ BMSC number did not change significantly (*Figure 1L*). Fetal bone marrow Nes-GFP⁺ cells thus include a small subset (<10%) of endothelial cells and a large population of non-endothelial stromal cells (>90%). Unlike Nes-GFP⁻ stromal cells, Nes-GFP⁺ cells proliferate slowly and do not express osteochondral protein cell markers.

## Bone marrow nestin⁺ cells do not contribute to fetal endochondrogenesis

We next studied whether Nes-GFP⁺ cells displayed osteoprogenitor activity in fetal bone marrow. The axial and appendicular skeleton is thought to originate solely from mesoderm. During endochondral ossification, cartilage is progressively replaced by osteoblast precursors that express the transcription factor osterix and infiltrate the perichondrium along the invading blood vessels (*Maes et al., 2010*). To identify mesodermal derivatives, we performed lineage-tracing studies by crossing mice expressing the *RCE* reporter—a sensitive reporter that drives stronger GFP expression than other reporter lines (*Sousa et al., 2009*)—with mice expressing inducible *Cre* recombinase under the regulatory elements of the *Hoxb6* gene, which is expressed in the lateral plate mesoderm (*Nguyen et al., 2009*). The resulting double-transgenic mice were administered tamoxifen at E10.5, a stage when the *Hoxb6* gene is still expressed. These mice and newborn *Nes-gfp* embryos were analyzed for osterix protein expression, which marks cells committed to the osteoblast lineage. Unlike osteoblast precursors derived from lateral plate mesoderm, Nes-GFP⁺ cells in fetal-limb bone marrow did not express highly osterix protein (*Figure 2A,B*).

We next performed genetically inducible fate mapping using Nes-*CreER^{T2}* mice (*Balordi and Fishell, 2007*). In these mice, tamoxifen administration triggers labeling of Nes-GFP⁺ cells and their progeny (*Figure 2C,D*). Tamoxifen was administered at E13.5 (when primary ossification centers start forming) (*Maes et al., 2010*), and at E8.5, to mark earlier nestin⁺ embryonic precursors. Unlike *Hoxb6*-traced mesodermal derivatives, nestin⁺ cells did not contribute to cartilage formation during this period. In contrast, *Nes*-traced cells with a similar morphology and distribution to Nes-GFP⁺ cells were observed in the chondro–osseous junction (*Figure 2E,F*). Similarly, Nes-GFP⁺ cells were not present inside the cartilage but were found in the innermost part of the perichondrium (*Figure 2—figure supplement 1A*), a region enriched in MSCs (*Maes et al., 2010*; *Yang et al., 2013*; *Zaidi and Mendez-Ferrer, 2013*). The results thus show that, unlike mesodermal derivatives, Nes-GFP⁺ cells do not exhibit osteochondral progenitor activity in the fetal bone marrow.

## MSC activity is progressively enriched in nestin⁺ cells

The lack of a contribution by nestin⁺ cells to fetal endochondrogenesis raised questions regarding their putative MSC properties in fetal bone marrow. We therefore measured mesenchymal progenitor activity in purified bone-marrow stromal subsets using the fibroblastic colony-forming unit (CFU-F) assay (*Friedenstein et al., 1970*) and the multipotent self-renewing sphere-forming assay (*Mendez-Ferrer et al., 2010*) (*Figure 2—figure supplement 1B*). BMSCs were isolated according to Nes-GFP

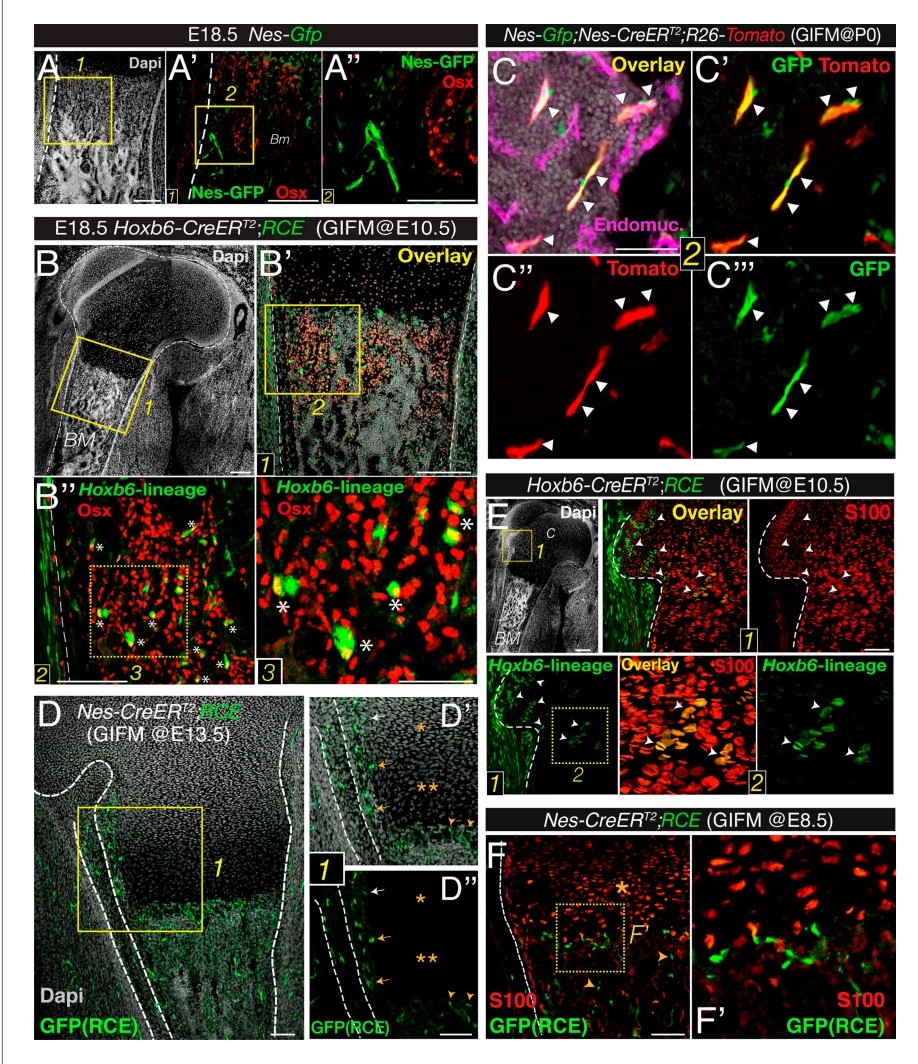

**Figure 2**. Bone marrow nestin+ cells are different from mesodermal osteo-chondroprogenitors. (**A** and **B**) Bone marrow sections from *Nes-Gfp* (**A–A''**) and *Hoxb6-CreER^T2;RCE* (**B–B''**) E18.5 embryos (tamoxifen-induced at E10.5) immunostained with Osterix antibodies (Osx, red) to label osteoprogenitor cells. GFP+ Osx+ mesodermal-derived osteoprogenitors are marked with asterisks (insets 2–3). (**C–C'''**) Perinatal recombination in *Nes-CreER^T2* mice efficiently targets bone marrow stromal Nes-GFP+ cells. Bone marrow section of a P7 *Nes-Gfp;Nes-CreER^T2;R26-Tomato* mouse that received tamoxifen at birth, showing Nes-GFP+ cells (green), *Nes*-derived progeny (red), and double-positive cells (arrowheads). (**D–F**) Fate mapping of the progeny of nestin+ cells and limb mesoderm in E18.5/19.5 femoral bone marrow from *Nes-CreER^T2;RCE* (**D–D''**, **F–F'**) and *Hoxb6-CreER;RCE* fetuses (**E**). (**D**) GFP (green) and nuclei counterstained with DAPI (gray) in bone of E18.5 fetus induced with tamoxifen at E13.5. Neither proliferating (*) nor hypertrophic (**) chondrocytes showed GFP fluorescence (inset 1). (**D'–D''**) *Nes*-derived cells with a similar morphology and distribution to Nes-GFP+ cells were detected near the cartilage–perichondrium interface (arrows) and within the chondro–osseous junction (arrowheads). (**E** and **F**) Bone marrow sections of (**E**) *Hoxb6-CreER;RCE* and (**F**) *Nes-CreER^T2;RCE* E18.5 embryos induced with tamoxifen at E10.5 and E8.5, respectively, stained with S100 antibodies to label chondrocytes (red). High magnification views of cartilage (inset 2) showing abundant double-positive chondrocytes (arrowheads). (**F**) *Nes*-traced cells (green) were not chondrocytes (red, *) but infiltrated the chondro–osseous junction and trabecular bone (arrowheads). Scale bars: 200 µm (**A–A'**, **B–B'**), 100 µm (**A''**, **B''**), 50 µm (**B''**3, **C**). *BM*, bone marrow; *C*, cartilage; *GIFM*, genetic inducible fate mapping.

The following figure supplement is available for figure 2:

**Figure supplement 1**. Sub-fractionation of fetal bone marrow mesenchymal progenitors.

expression (*Figure 2—figure supplement 1C*). CFU-F efficiency was nearly three times higher in Nes-GFP⁻ cells than in Nes-GFP⁺ cells at E17.5 (*Figure 3A*). Conversely, non-adherent sphere formation was markedly enriched in GFP⁺ cells, whereas most spheres derived from GFP⁻ cells rapidly attached to plastic and spontaneously differentiated into adipocytes (*Figure 3B–G*), suggesting that Nes-GFP⁻ BMSCs are in a more committed state. Spheres formed by Nes-GFP⁺ bone marrow cells contained mesenchyme-like spindle-shaped GFP⁺ cells (*Figure 3E*). At E18.5 and during the first postnatal week, CFU-F frequency was 6-fold higher in the GFP⁻ stromal population than in GFP⁺ cells (*Figure 3H*). However, at later postnatal stages, CFU-F activity was progressively restricted to Nes-GFP⁺ cells due to a sharp drop in activity in GFP⁻ BMSCs (>100-fold reduction between E18.5 and P14, compared with a 0.5-fold reduction in Nes-GFP⁺ cells). At P7, CFU-Fs derived from Nes-GFP⁻ cells contained mostly preosteoblasts (*Figure 3I–K*). The expression of genes associated with chondrocyte development was higher in Nes-GFP⁻ than in Nes-GFP⁺ BMSCs at E18.5; in contrast, the expression of master regulators of chondrogenesis, osteogenesis, and adipogenesis was progressively enriched in postnatal Nes-GFP⁺ BMSCs (*Figure 3L*), consistent with the increasing MSC enrichment in this population. Together, these results suggest that most fetal BMSCs do not express nestin and quickly differentiate towards committed skeletal precursors, losing most MSC activity by the second week after birth. In contrast, nestin⁺ cells conserve MSC activity throughout life.

## The trunk neural crest contributes to bone marrow nestin⁺ MSCs

Neural crest cells are characterized by nestin expression and sphere-forming ability. Although cells traced to neural crest origin have been reported in adult murine bone marrow (*Nagoshi et al., 2008*; *Morikawa et al., 2009b*; *Glejzer et al., 2011*; *Komada et al., 2012*), their precise identity, developmental dynamics, and function have remained elusive. Moreover, the neural crest-specific *Wnt1-Cre* line used in these studies displays ectopic *Wnt1* activation. To trace neural crest derivatives, we performed genetic fate-mapping studies with a recent *Wnt1-Cre2* line that does not induce ectopic Wnt1 activity (*Lewis et al., 2013*). Unexpectedly, limb bones from *Wnt1-Cre2;R26-Tomato* double-transgenic neonates showed some neural crest-derived osteoblasts and osteocytes aligning the most recent layers of bone deposition, as well as similarly distributed chondrocytes in the outermost layers of the femur head (*Figure 4A,B*). As expected, neural crest-traced Schwann cells expressing glial fibrillary acidic protein (GFAP) were also detected in the bone marrow of one-week old mice (*Figure 4—figure supplement 1A,B*). Intriguingly, GFAP⁻ perivascular cells with a similar morphology and distribution to Nes-GFP⁺ cells were also derived from Wnt1⁺ cells (*Figure 4—figure supplement 1C,D*). The number of neural crest-traced osteochondral cells increased in the first postnatal week (*Figure 4C*). By P28, CFU-F activity was much higher in *Wnt1-Cre2*-traced cells than in non-neural crest-traced bone marrow stromal cells (*Figure 4D*). These results show that the neural crest contributes to limb bones late in development.

We further characterized cell surface marker expression by neural crest-derived MSCs. Platelet-derived growth factor receptor alpha (Pdgfrα) is required in mesodermal and neural crest-derived mesenchyme during development (*Schatteman et al., 1992*; *Soriano, 1997*). Mouse Pdgfrα⁺ BMSCs are highly enriched in CFU-F activity (*Takashima et al., 2007*; *Morikawa et al., 2009a*) and most adult mouse bone marrow nestin⁺ cells are also Pdgfrα⁺ (*Yamazaki et al., 2011*; *Pinho et al., 2013*). We found that Pdgfrα⁺ BMSCs were also enriched in CFU-F activity at fetal stages (*Figure 4—figure supplement 1E*). In the bone marrow of 4-week old *Nes-Gfp;Wnt1-Cre2;R26-Tomato* mice, most neural crest-traced cells were also Pdgfrα⁺ and Nes-GFP⁺ (*Figure 4E*). For confirmation, we intercrossed *Nes-Gfp;R26-Tomato* mice with a line expressing tamoxifen inducible Cre recombinase under the regulatory elements of the gene encoding the neural crest transcription factor Sox10. *Nes-Gfp;Sox10-CreER^{T2};R26-Tomato* mice were administered tamoxifen at E9.5 to label migratory neural crest-derived cells. Similar to the situation in stage-matched *Nes-Gfp* mice—and also consistent with *Wnt1-Cre2*-traced cells—most *Sox10-CreER^{T2}*-traced bone marrow stromal cells were Pdgfrα⁺ and Nes-GFP⁺ cells (*Figure 4F* and *Figure 4—figure supplement 2*). These results thus demonstrate definitively that the neural crest contributes to nestin⁺ BMSCs.

## The Nes-GFP⁺ bone marrow population comprises Pdgfrα⁺ MSCs and Pdgfrα⁻ Schwann cell precursors

The finding that some fetal bone marrow Nes-GFP⁺ cells expressed Pdgfrα while others did not prompted us to study the possible functional heterogeneity of this population. Recent work showed

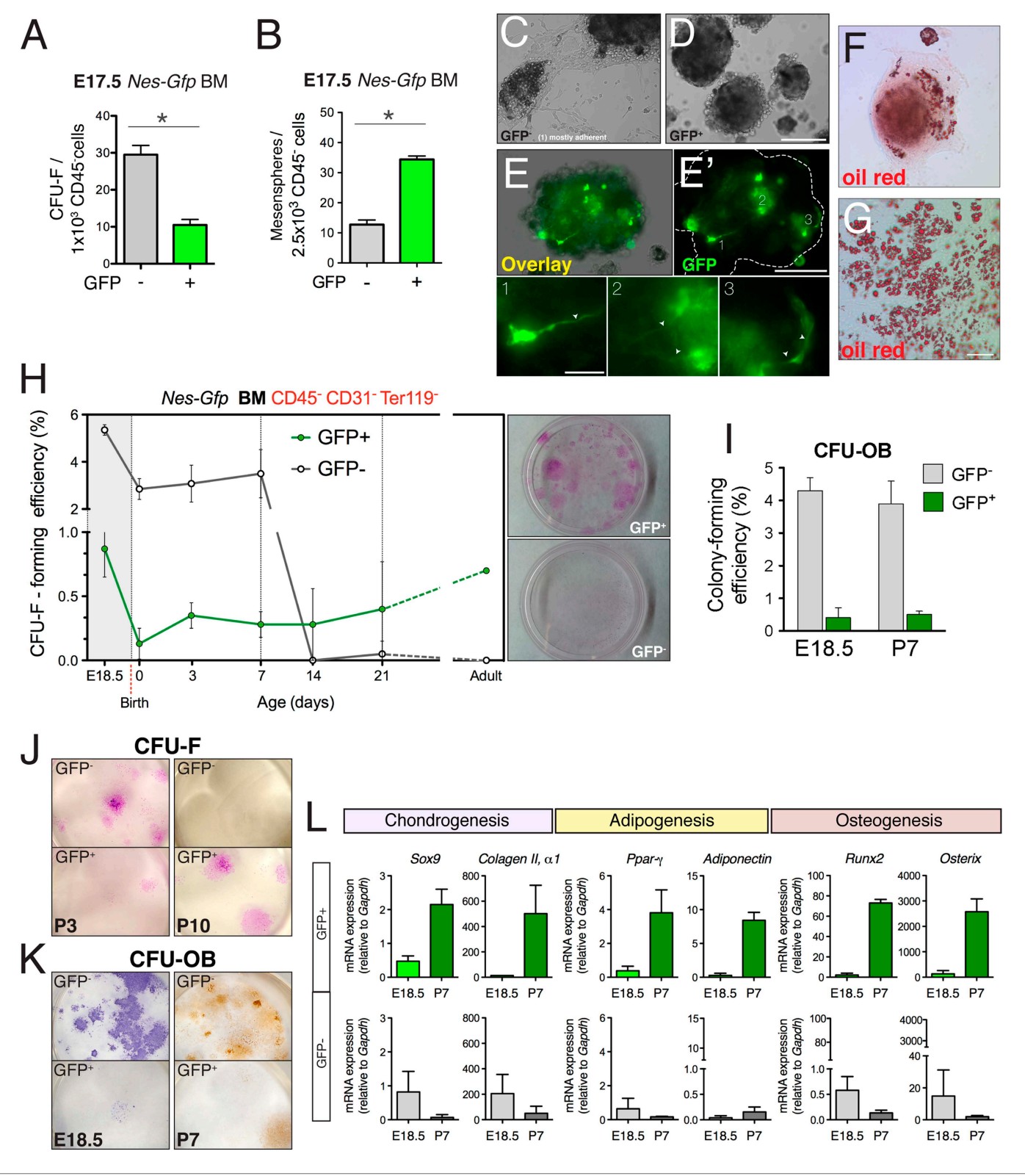

**Figure 3**. Perinatal enrichment of MSC activity in bone marrow nestin+ cells. (**A** and **B**) Fibroblast colony-forming units (CFU-F) and mesensphere-forming activities segregate in Nes-GFP- and Nes-GFP+ fetal bone marrow cells, respectively. Frequencies of CFU-F and mesensphere-forming efficiency in E17.5 *Nes-Gfp* embryos. (**C–E**) Representative sphere cultures from both mesenchymal subpopulations sorted from bone marrow of *Nes-Gfp* fetuses. Note the
*Figure 3. Continued on next page*

*Figure 3. Continued*

presence of GFP+ fibroblast-like cells (**E'** and insets1–3). (**F–G**) Adherent colonies derived from GFP- population stained with Oil Red O (red), to reveal mature adipocytes. (**H**) MSC activity is progressively restricted to bone marrow Nes-GFP+ cells. Frequency of CFU-Fs in cultures of stromal (CD45− CD31− Ter119−) GFP+/− cells, isolated from the bone marrow of *Nes-Gfp* mice of the indicated age. Right panels show representative CFU-Fs in cell populations from adult mice. (**I**) Frequency of osteoblastic colony-forming units (CFU-OB) in bone marrow stromal GFP+/− cells of the indicated age. (**J**) Representative Giemsa-stained CFU-F from 3- and 10-day old bone marrow subpopulations. (**K**) Stained CFU-OB from E18.5 (alkaline phosphatase staining, left panels) and 1-week old (alizarin red staining, right panels) bone marrow subpopulations. (**L**) qPCR analysis of mesenchymal genes in bone marrow stromal populations isolated from fetal (E18.5) or 1-week old (P7) *Nes-Gfp* mice, as depicted (Figure 3—figure supplement 1). (**A–B**, **H–L**) Mean ± SD, n = 3–6; *p < 0.05, unpaired two-tailed *t* test. Scale bars: 200 μm (**D**, **E'**, **G**), 100 μm (**G**), 50 μm (**E'**1–3).

that most adult bone marrow nestin+ cells are Pdgfrα+ and also that nestin+ Pdgfrα+ Schwann cells contribute to HSC maintenance (*Yamazaki et al., 2011*). We found that bone marrow Nes-GFP+ cells were closely associated with distinctive Gfap+ Schwann cells (*Figure 4—figure supplement 1G*). After sorting of neonatal GFP+/− Pdgfrα+/− BMSCs (*Figure 5A*), the two populations were analyzed by next-generation sequencing. Detection of endogenous *Pdgfrα* and *Nes* transcripts verified the isolation strategy. Interestingly, whereas *Ly6a/Sca1* expression was higher in GFP−Pdgfrα+ cells (*Figure 5—figure supplement 1A,B*), the expression levels of HSC maintenance genes (*Cxcl12*, *Kitl* and *Angpt1*) and the *Leptin receptor*, which marks HSC niche-forming mesenchymal cells (*Ding et al., 2012*), was highly enriched in GFP+ Pdgfrα+ cells (*Figure 5B*). This population also abundantly expressed other genes enriched in MSCs (*Figure 5—figure supplement 1C*). In contrast, Nes-GFP+ Pdgfrα− cells expressed genes characteristic of Schwann cell precursors (*Sox10*, *Plp1*, *Erbb3*, *Dhh*) but did not express mature Schwann cell genes, such as *Gfap* (*Figure 5C*). Gene ontology analysis of differentially expressed genes between the two Pdgfrα+ subpopulations revealed enrichment of categories related to ossification, bone and blood vessel development, axon guidance, and Schwann cell differentiation (*Figure 5—figure supplement 1D and supplement 2*).

To further characterize nestin+ subpopulations, we compared the transcriptome-wide profile of neonatal Nes-GFP+/− Pdgfrα+/− BMSCs with publicly available microarray expression data sets from primary adult BMSCs or neural crest derivatives (*Table 1*). Unbiased hierarchical clustering and principal component analysis (*Figure 5D*) revealed that Nes-GFP+ Pdgfrα+ cells were more similar to adult primitive BMSCs and distinct from more differentiated osteoblastic cells (*Nakamura et al., 2010*). Pdgfrα+ Nes-GFP+/− cells clustered nearby, consistent with the increasing restriction of Pdgfrα to Nes-GFP+ cells in postnatal bone marrow. In addition, Nes-GFP+ Pdgfrα + cells clustered far from Nes-GFP+ Pdgfrα− cells, whose genomic profile was closest to that of E12.5 Schwann cell precursors. MSC-like and neural crest stem-cell-like derived clones (*Wislet-Gendebien et al., 2012*) were markedly different, probably because these were cultured cells. Intriguingly, we noted a maturation hierarchy of Schwann and osteolineage cells, from undifferentiated cells (*Figure 5D*, lower corners) to more mature lineages (*Figure 5D*, contralateral upper corners). At the intersection of these differentiation waves, adult bone marrow CD45− Nes-GFP+ cells (*Mendez-Ferrer et al., 2010*) converged with bone marrow HSC niche cells identified by the expression of stem cell factor (*Ding et al., 2012*). These results suggest the existence of two nestin+ populations with non-overlapping MSC and Schwann cell precursor features. To test this hypothesis functionally, we cultured neonatal Nes-GFP+/− Pdgfrα+/− BMSCs in differentiation medium, finding that mesenchymal and glial differentiation was segregated in Pdgfrα+ and Pdgfrα− cells, respectively (*Figure 5E,F*). Thus two Nes-GFP+ neural crest derivatives occur in postnatal bone marrow: Pdgfrα+ MSCs enriched in HSC-supporting genes and Pdgfrα− Schwann cell precursors.

## Deficient migration of neural crest-derived cells along nerves reduces the bone marrow MSC and HSC populations

Under similar culture conditions used to grow neural crest cells, adult mouse bone marrow Nes-GFP+ cells can form self-renewing and multipotent mesenchymal spheres with the capacity to transfer hematopoietic activity to ectopic sites during serial transplantations (*Mendez-Ferrer et al., 2010*). In addition, human bone marrow-derived mesenspheres secrete factors that can expand human cord blood HSCs through secreted factors (*Isern et al., 2013*). These findings and the results presented so far together suggest that neural crest-derived MSCs might have a specialized function in establishing the HSC niche in the developing bone marrow. We further studied the role of neural crest-derived cells in

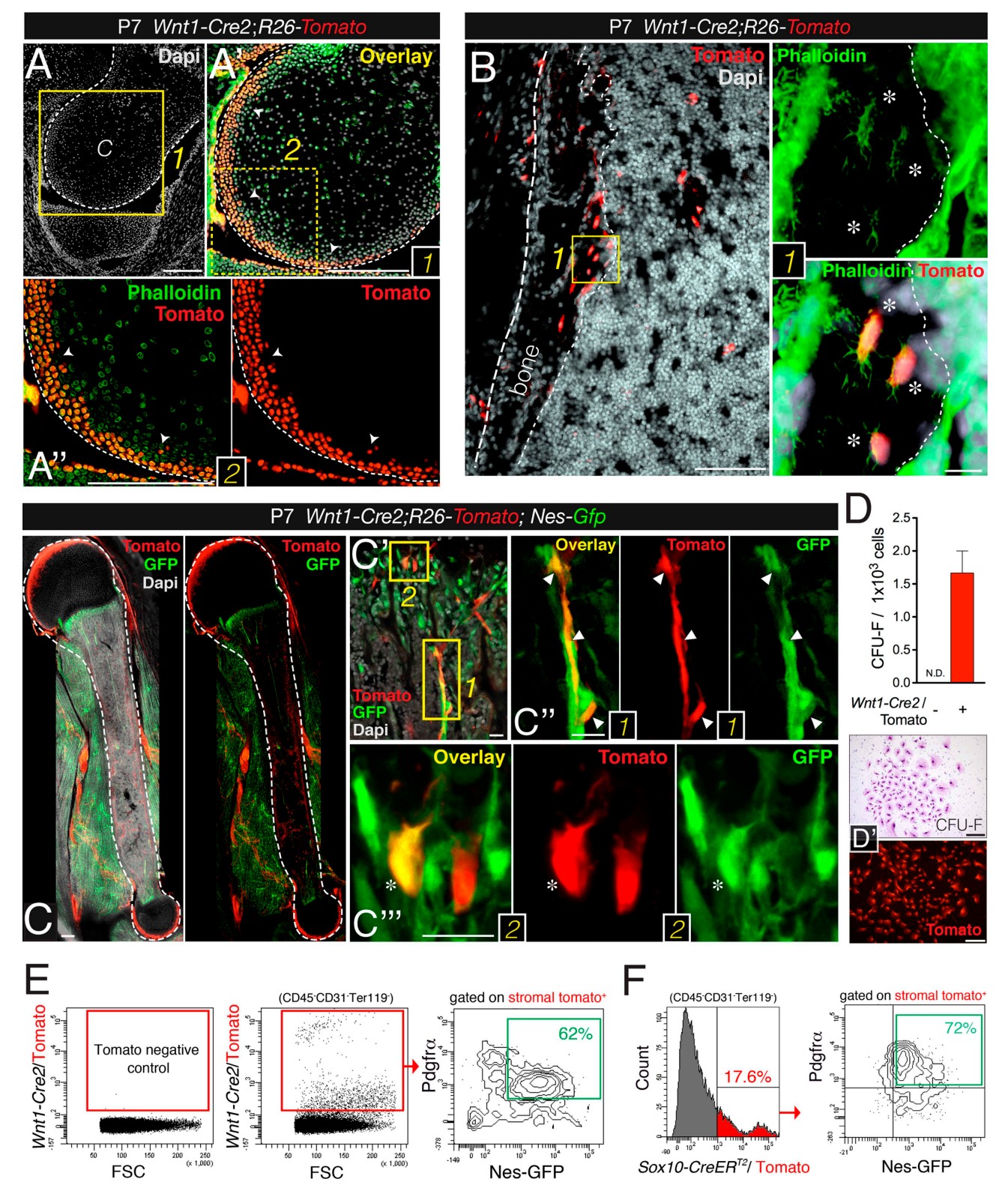

**Figure 4.** Contribution of trunk neural crest cells to mesenchymal lineages in long bones. (**A** and **B**) Fate mapping of neural crest derivatives in femoral bone marrow of neonatal *Wnt1-Cre2;R26-Tomato* mice. (**A**) Section through femoral distal epiphysis showing cortical neural-crest-derived chondrocytes (arrowheads, red); green signal corresponds to phalloidin staining. Nuclei were counterstained with DAPI (gray). (**B**) Bone marrow section showing

*Figure 4. Continued on next page*

*Figure 4. Continued*

*Wnt1-Cre2*-derived Tomato+ (red) osteocytes. (Inset 1) Neural-crest-derived osteocytes (asterisks) in endosteal region, showing their typical morphology revealed by phalloidin staining (green). (**C–E**) The neural crest contributes to Pdgfrα+ BMSCs in long bones. (**C–C''**) Fluorescent signals of GFP, Tomato, and DAPI in bone marrow sections from 1-week old *Wnt1-Cre2;R26-Tomato;Nes-Gfp* mice. (**D**) Frequency of fibroblastic colony-forming units (CFU-F) in CD31- CD45- Ter119- Tomato+/- bone marrow cells sorted from 1-week old *Wnt1-Cre2;R26-Tomato* mice (*n* = 3); N.D., not detectable. (**D'**) Examples of Giemsa staining (top panel) and Tomato fluorescence in neural crest-derived CFU-Fs. (**E**) Representative flow cytometry analysis of bone marrow stromal cells from 4-week old *Wnt1-Cre2;R26-Tomato;Nes-Gfp* mice. (**F**) Flow cytometry analysis of bone marrow stromal cells from *Nes-Gfp;Sox10-CreER^{T2};R26-Tomato* triple-transgenic mice stained with Pdgfrα antibody. (**E, F**) Frequencies of neural crest-traced BMSCs are indicated. Scale bars: 200 μm (**A–A''**, **C**, **D–D'**), 100 μm (**B**), 20 μm (**B1**, **C'–C'''**). Dashed line depicts the bone and cartilage contour (**A–C**).

The following figure supplements are available for figure 4:

**Figure supplement 1**. Bone marrow Nes-GFP+ cells are distinct from mature Schwann cells.

**Figure supplement 2**. Contribution of trunk neural crest to bone marrow stromal lineages.

this process using a loss-of-function model. Perineural migration of neural crest-derived cells requires the interaction of the receptor tyrosine-protein kinase ErbB3 with the ligand neuregulin-1, produced by developing nerves (*Jessen and Mirsky, 2005*). *Erbb3*-deficient mice initially show normal development of peripheral nerves but later display impaired perineural migration of neural crest-derived cells and die at perinatal stage (*Riethmacher et al., 1997*). Hematopoietic progenitors were increased in fetal liver of KO mice (*Figure 6A*). In contrast, expression of the MSC marker CD90, enriched in Nes-GFP+ cells (*Figure 6B,C*), was reduced two-fold in *Erbb3^{-/-}* limb bone marrow, associated with 5-fold drop in the number of bone marrow hematopoietic progenitors (*Figure 6D,G*). To further dissect the contribution of neural crest to fetal hematopoiesis, we performed a similar analysis in mice conditionally lacking ErbB3 in Schwann-committed cells. To label Schwann cells, we intercrossed *R26-Tomato* reporter mice intercrossed with a line expressing Cre recombinase under the regulatory elements of desert hedgehog (*Dhh*) promoter (*Jaegle et al., 2003*) (*Figure 6—figure supplement 1A–C*). These *Dhh-Cre* mice were then intercrossed with the ErbB3 conditional KO. Similar to the constitutive KO, *Dhh-Cre;Erbb3^{fl/fl}* mice are virtually devoid of Schwann cells (*Sheean et al., 2014*); however, unlike the constitutive KO, *Dhh-Cre;Erbb3^{fl/fl}* mice had a normal frequency of bone marrow hematopoietic progenitors (*Figure 6—figure supplement 1D*). Together, these results suggest that neural crest cells not yet committed to the Schwann cell lineage migrate along developing nerves to the bone marrow, giving rise to HSC niche-forming MSCs.

## Neural crest-derived nestin+ cells direct developmental HSC migration to the bone marrow

Despite the low frequency of HSCs and Nes-GFP+ cells, detailed immunofluorescence analysis showed significant proximity of HSCs to Nes-GFP+ cells in neonatal bone marrow (*Figure 7A,B*), suggesting that nestin+ cells might attract circulating HSCs toward their final developmental destination in the bone marrow. To study the contribution of nestin+ cells to HSC migration from fetal liver to bone marrow, we used mice expressing the diphtheria toxin (iDTA) or its receptor (iDTR) in nestin+ cells. Depletion of nestin+ cells at E15.5 in *Nes-CreER^{T2};iDTR* mice caused an ~4-fold reduction in fetal bone marrow HSC activity within 48 hr, inversely correlating with an ~8-fold increase in fetal liver HSC activity (*Figure 7C,D*). The cell cycle profile and apoptosis in hematopoietic progenitors were unchanged (*Figure 7—figure supplement 1A*), but their numbers increased in fetal liver by 40% (*Figure 7E*). Similar results were obtained by depleting nestin+ cells during the first postnatal week in *Nes-CreER^{T2};iDTA* mice, which otherwise showed normal bone marrow histology (*Figure 7F* and *Figure 7—figure supplement 1B*). Developmental HSC migration to bone marrow proceeds until the second week after birth (*Dzierzak and Speck, 2008*), suggesting that the bone marrow environment might still mature during this period to accommodate HSCs. We therefore analyzed GFP+/- BMSCs from E18.5 and P7 *Nes-Gfp* bone marrow. The expression of HSC-supporting genes was markedly higher and progressively upregulated in GFP+ cells during the first postnatal week (*Figure 7G*). These results suggest that perinatal maturation of nestin+ cells allows the colonization of bone marrow by circulating HSCs.

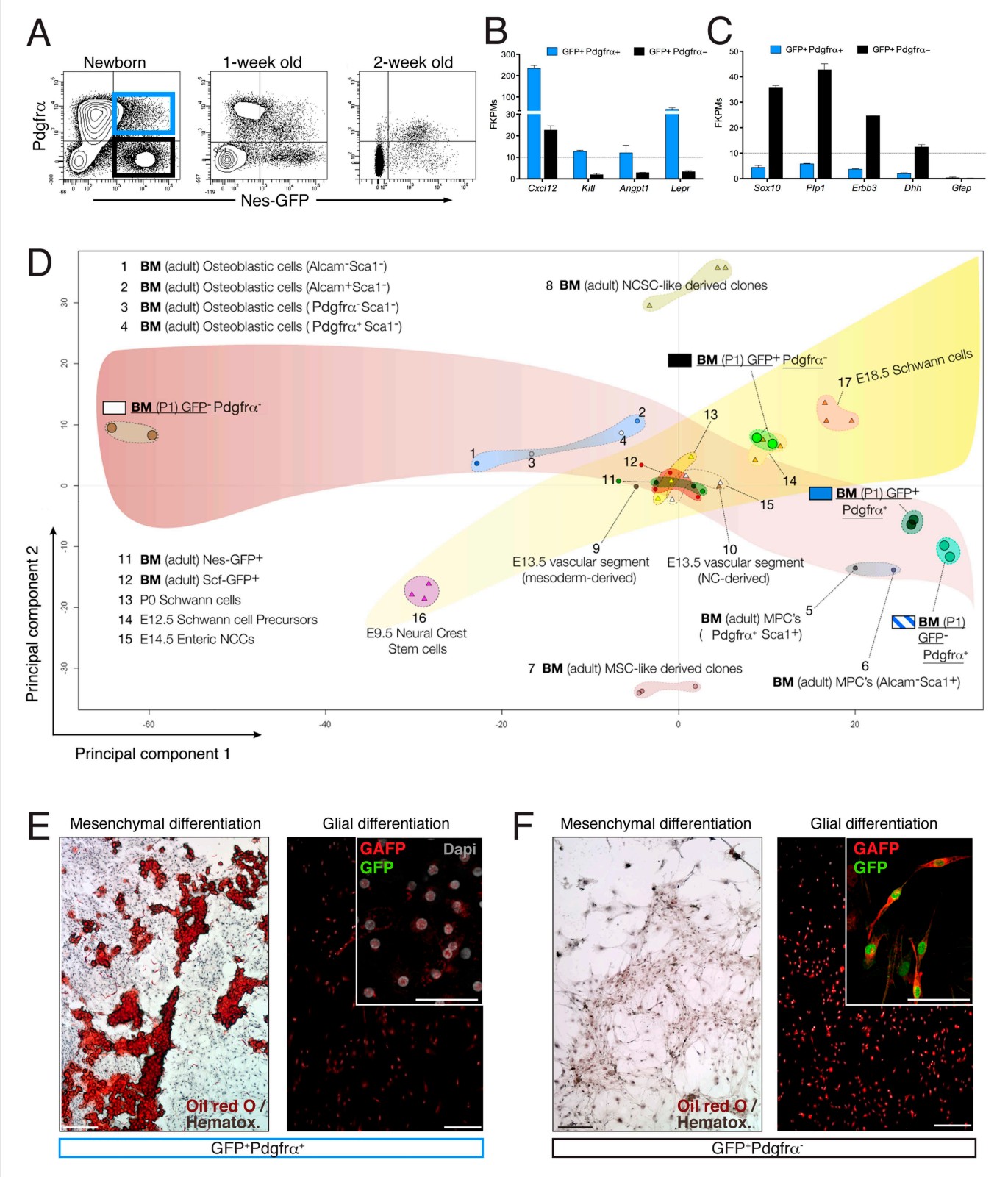

**Figure 5**. The neonatal bone marrow Nestin-GFP+ population contains Pdgfrα+ MSCs and Pdgfrα− Schwann cell precursors. (**A**) Representative flow cytometry profiles showing Nes-GFP and Pdgfrα expression in postnatal BMSCs. (**B** and **C**) Relative mRNA expression levels of (**B**) HSC niche-related genes and (**C**) Schwann cell progenitor genes by GFP+ Pdgfrα+ (black) or GFP+ Pdgfrα− BMSCs. RNAseq data are expressed as fragments per kilobase of

*Figure 5. Continued on next page*

*Figure 5. Continued*

exon per million fragments mapped (FPKM; *n=2* independent samples from pooled newborns). Note that the neonatal GFP⁺ Pdgfrα⁻ subpopulation has a Schwann cell progenitor signature (**C**), whereas GFP⁺ Pdgfrα⁺ cells are enriched in HSC maintenance genes. (**D**) Principal component analysis comparing the transcriptome of neonatal *Nes-Gfp* bone marrow stromal subsets with available microarray expression data sets from neural crest-derived populations and primary adult mouse BMSCs (*Table 1*). (**E** and **F**) In vitro differentiation of neonatal subpopulations isolated as in (**A**) and cultured in mesenchymal (mesenchymal) and Schwann cell (glial) differentiation medium. Adipocytes were stained with Oil Red O (red) and counterstained with hematoxylin (left panels); Schwann cells were stained with antibodies against glial fibrillary acidic protein (Gfap, red) and overlaid with endogenous GFP fluorescence (right panels). Scale bars: 200 μm (top right insets: 50 μm).

The following figure supplements are available for figure 5:

**Figure supplement 1**. The neonatal Nes-GFP⁺ bone marrow population is enriched in primitive mesenchymal progenitors.

**Figure supplement 2**. RNA-seq data analysis.

## Nestin⁺ MSC-derived Cxcl12 contributes to the establishment of the HSC niche in the bone marrow

HSC migration to fetal bone marrow is enhanced by Cxcl12 and stem cell factor (*Christensen et al., 2004*), both of which are highly expressed and progressively upregulated in bone marrow nestin⁺ cells at perinatal stages (*Figure 7G*). Cxcl12 is produced by several stromal cells and is required for developmental bone marrow colonization by HSCs (*Ara et al., 2003*). It has been argued that Cxcl12 produced by endothelial cells and nestin⁻ mesenchymal progenitors—but not by nestin⁺ cells—is necessary for adult HSC maintenance (*Ding and Morrison, 2013*; *Greenbaum et al., 2013*). We found that, one week after birth, *Cxcl12* mRNA levels in Nes-GFP⁺ BMSCs were >20-fold higher than in bone marrow endothelial cells and 80-fold higher than in Nes-GFP⁻ BMSCs (*Figure 7H*). Among neural crest-traced cells, Nes-GFP⁺ BMSCs were particularly enriched in the expression of *Cxcl12* and endogenous *Nestin* (*Figure 7I,J*). To conditionally delete Cxcl12 in nestin⁺ cells in the first post-natal week, we intercrossed *Cxcl12* fl mice (*Tzeng et al., 2010*) with *Nes-CreER*T2 mice, which mostly label Nes-GFP⁺ cells during this period (*Figure 7K*). Tamoxifen administration did not significantly alter *Cxcl12* mRNA levels in bone marrow endothelial cells but decreased these levels by 5-fold in BMSCs (*Figure 7L–M*). This was associated with an ~30% reduction of bone marrow hematopoietic progenitors and HSCs measured by long-term competitive repopulation assays (*Figure 7N,O*). These results demonstrate that Cxcl12 production by nestin⁺ MSCs contributes to the HSC niche formation in the developing bone marrow.

## Discussion

The aim of this study was to investigate the ontogeny and specific functions of mesenchymal progenitors in the fetal bone marrow. We show that the developing bone marrow in axial and appendicular skeleton harbors different MSC populations with distinct origins and specialized roles. While mesoderm-derived nestin⁻ MSCs give rise to bone and cartilage, the neuroectoderm provides an additional source of MSCs, marked by nestin expression, that are endowed with specific HSC niche functions. We therefore conclude that osteochondroprogenitor and stem cell niche functions are separate and non-overlapping during bone marrow ontogenesis. The neural crest thus gives rise to three regulators of adult HSC activity: sympathetic neurons, associated Schwann cells, and nestin⁺ MSCs.

Although it is accepted that mammalian connective tissues, such as bone or skeletal muscle, are derived mainly from mesoderm, the precise origin of BMSCs has remained unclear. While the neural crest contributes to the craniofacial skeleton, the trunk neural crest is thought to generate mostly non-ectomesenchymal derivatives, including melanocytes, neurons, and glia of the peripheral nervous system. In the trunk skeleton, mesenchymal cells have thus been considered to be derived mostly from the mesoderm (*Olsen et al., 2000*), but the neural crest is also a source of pericytes, mural cells, and fibroblasts (*Bergwerff et al., 1998*; *Etchevers et al., 2001*; *Joseph et al., 2004*) that can differentiate into mesenchymal lineages in vitro (*Morikawa et al., 2009b*; *Glejzer et al., 2011*; *John et al., 2011*; *Komada et al., 2012*). In addition, mature endothelial cells can generate mesenchymal cells through endothelial-to-mesenchymal transition. Genetic fate-mapping studies using the neuroepithelial marker *Sox1* identified a neuroectodermal origin of the earliest

**Table 1.** Description of publicly available data sets used for principal component analyses

| Plot ID | Cell Population | GEO* samples | Sample description | Ref |
|---|---|---|---|---|
| 1 | BM osteoblastic cells (Alcam⁻ Sca1⁻) | GSM437794 | BM (adult) primary stromal† Alcam⁻ Sca1⁻ | ‡ |
| 2 | BM osteoblastic cells (Alcam⁺ Sca1⁻) | GSM437795 | BM (adult) primary stromal† Alcam⁺ Sca1⁻ | |
| 3 | BM osteoblastic cells (Pdgfrα⁻ Sca1⁻) | GSM437797 | BM (adult) primary stromal† Pdgfrα⁻ Sca1⁻ | |
| 4 | BM osteoblastic cells ( Pdgfrα⁺ Sca1⁻) | GSM437798 | BM (adult) primary stromal† Pdgfrα⁺ Sca1⁻ | |
| 5 | BM MPC's (Pdgfrα⁺ Sca1⁺) | GSM437799 | BM (adult) primary stromal† Mesenchymal progenitor cells (MPC) (Pdgfrα⁺ Sca1⁺) | |
| 6 | BM MPC's (Alcam⁺ Sca1⁺) | GSM437796 | BM (adult) primary stromal† Mesenchymal progenitor cells (MPC) (Alcam⁺ Sca1⁺) | |
| 7 | BM MSC-like derived clones | GSM795638-40 | BM MSC-derived cell line, Wnt1Cre/R26R β-gal⁻ selected clone passage>10 | § |
| 8 | BM NCSC-derived clones | GSM795641-43 | BM NCSC-derived cell line, Wnt1Cre/R26R β-gal⁻ selected clone passage>10 | |
| 9 | E13.5 vascular segment (mesoderm-derived) | GSM261911 | E13.5 internal carotid artery vascular segment (smooth muscle mesoderm-derived) | ¶ |
| 10 | E13.5 vascular segment (NC derived) | GSM261912 | E13.5 external carotid artery vascular segment (smooth muscle NC-derived) | |
| 11 | BM (adult) Nes-GFP⁺ | GSM545815-17 | BM (adult) primary (CD45⁻) Nes-GFP⁺ cells | ** |
| 12 | BM (adult) Scf-GFP⁺ | GSM821066-68 | BM (adult) primary Scf-GFP+ cells | †† |
| 13 | P0 Schwann cells | GSM15386-88 | P0 primary Schwann cells (Plp-GFP⁺) from sciatic nerve | ‡‡ |
| 14 | E12.5 Schwann cell precursors (SCPs) | GSM15373-75 | E12.5 primary Schwann cell precursors (Plp-GFP⁺) from sciatic nerve | |
| 15 | E14.5 Enteric Neural crest cells (ENCC) | GSM844492-94 | E14.5 primary ENCCs (Wnt1Cre/R26-YFP⁺) from gut | |
| 16 | E9.5 Neural crest stem cells (NCSC) | GSM15370-72 | E9.5 trunk primary Plp-GFP+ cells (migrating NCSCs) | |
| 17 | E18.5 Schwann cells | GSM15383-85 | E18.5 primary Schwann cells (Plp-GFP⁺) cells from sciatic nerve | |

*Gene Expression Omnibus database (http://www.ncbi.nlm.nih.gov/geo/)

†CD45⁻CD31⁻Ter119⁻

**References:**

‡Nakamura *et al.* Isolation and characterization of endosteal niche cell populations that regulate hematopoietic stem cells. Blood (2010) vol. 116 (9) pp. 1422-32.

§Wislet-Gendebien *et al.* Mesenchymal stem cells and neural crest stem cells from adult bone marrow: characterization of their surprising similarities and differences. Cell Mol Life Sci (2012)vol. 69 (15) pp. 2593-608.

¶Zhang *et al.* Origin-specific epigenetic program correlates with vascular bed-specific differences in Rgs5 expression. FASEB J (2012) vol. 26 (1) pp. 181-91.

**Méndez-Ferrer *et al.* Mesenchymal and haematopoietic stem cells form a unique bone marrow niche. Nature (2010) vol. 466 (7308) pp. 829-34.

††Ding and Morrison. Haematopoietic stem cells and early lymphoid progenitors occupy distinct bone marrow niches. Nature (2013) pp. 1-6.

‡‡Buchstaller *et al.* Efficient isolation and gene expression profiling of small numbers of neural crest stem cells and developing Schwann cells. J Neurosci (2004) vol. 24 (10) pp. 2357-65.

trunk MSCs, but other MSCs are recruited from undefined sources at later stages (*Takashima et al., 2007*). This picture raised the question of whether several MSCs might transiently coexist in the developing bone marrow and whether neural-crest-derived MSCs with specific functions might persist in the postnatal bone marrow.

In this study, we show that most mesenchymal activity and chondrogenic capacity in the fetal bone marrow is associated with nestin⁻ MSCs, but that these rapidly differentiate towards committed osteochondral lineages early in postnatal life. In contrast, slow-proliferating neural crest-derived nestin⁺ MSCs do not contribute to fetal endochondrogenesis but are instead required to establish the HSC niche in the same bones. Interestingly, we also found that nestin⁺ cells retained

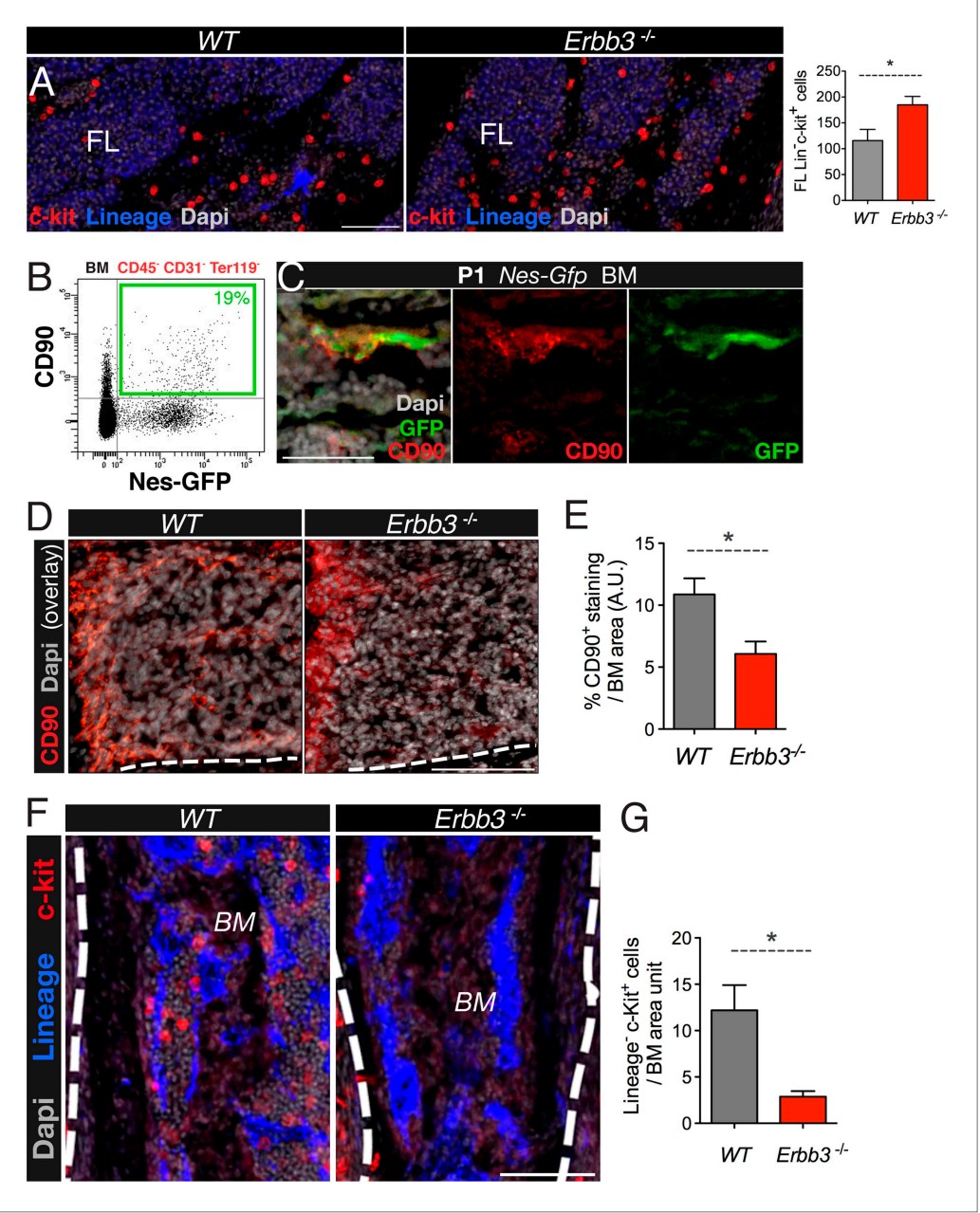

**Figure 6**. Perineural migration of neural crest-derived cells to long bones generates nestin⁺ MSCs with specialized HSC niche function. (**A**) E17.5-E18.5 fetal liver (FL) sections from wild-type (wt) and *Erbb3*-null embryos stained with antibodies for mature hematopoietic lineage (blue) and c-Kit (red). Quantification of fetal liver Lin⁻ c-Kit⁺ hematopoietic progenitors (per 0.41 mm²). (**B**) Representative FACS profile of CD45⁻ CD31⁻Ter119⁻ bone marrow cells from 2-week old *Nes-Gfp* mice stained with the mesenchymal marker CD90, showing the expression enrichment in Nes-GFP⁺ cells. (**C**) Neonatal bone marrow section stained with anti-CD90 (red), which labeled Nes-GFP⁺ (green) cells. Scale bar: 50 μm. (**D**) Representative bone marrow sections from wt and *Erbb3*-null E17.5/18.5 mice immunostained with anti-CD90 (red). (**E**) Quantification of CD90 immunostaining of samples in (**D**); *n* = 3. (**F**) Staining of bone marrow sections from wt and *Erbb3*-null embryos with antibodies for mature hematopoietic lineage (blue) and c-Kit (red). (**G**) Quantification of bone marrow Lineage⁻ c-Kit⁺ hematopoietic progenitors in E17.5/18.5 wt and *Erbb3*-null mice (*n* = 3). (**E**, **G**) Mean ± SEM; *p < 0.05, unpaired two-tailed *t* test.

The following figure supplement is available for figure 6:

**Figure supplement 1**. Conditional *Erbb3* deletion after glial specification of Schwann cell precursors does not affect bone marrow HSCs.

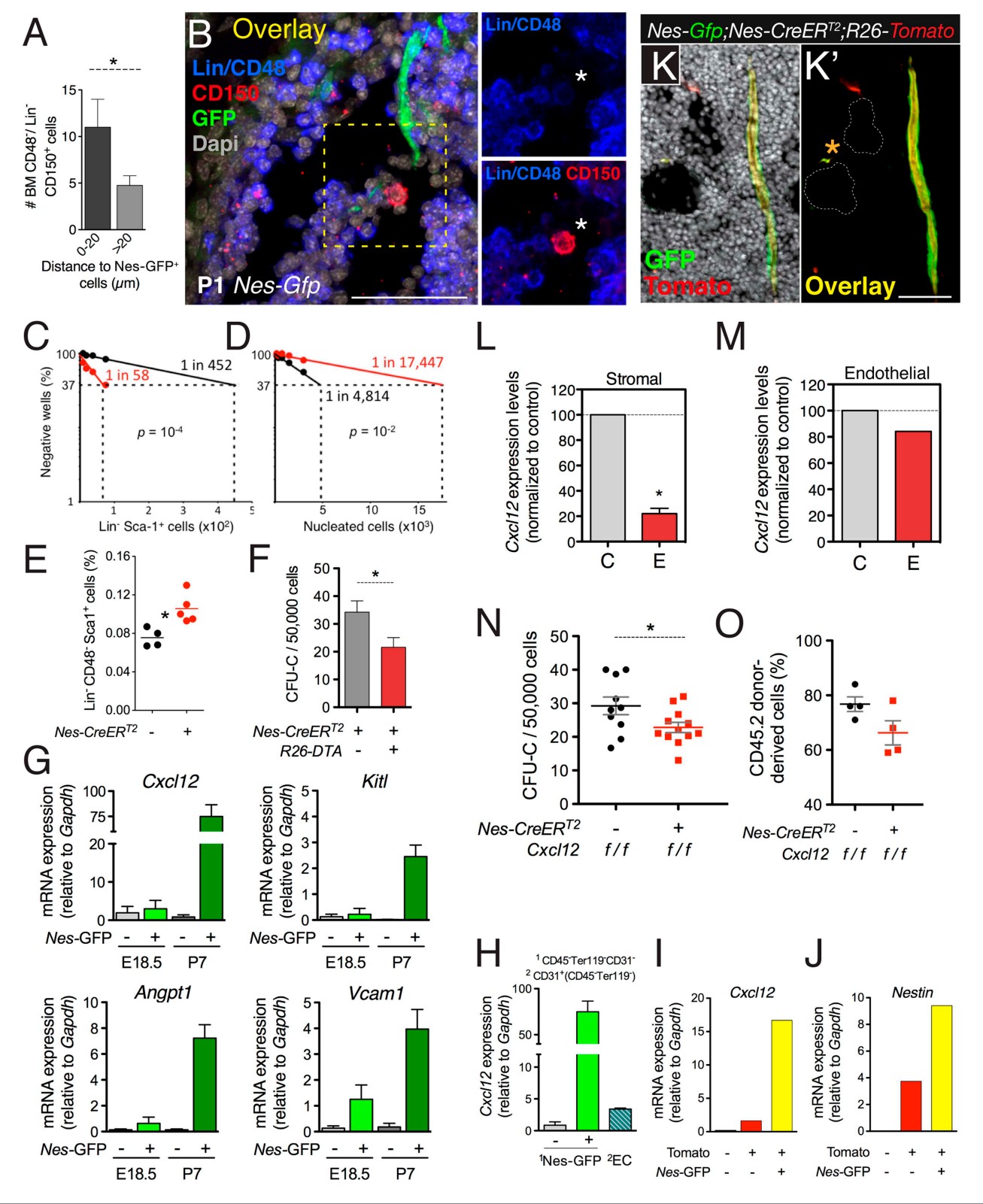

**Figure 7**. CXCL12 produced by nestin[+] MSCs contributes to the establishment of the HSC niche in the bone marrow. (**A** and **B**) HSCs are localized near Nes-GFP[+] cells in neonatal bone marrow. Neonatal femoral sections from *Nes-Gfp* mice were immunostained with antibodies for mature hematopoietic lineages, CD48 (blue) and CD150 (red). (**A**) Quantification of the distance of Lin[−] CD48[−] CD150[+] HSC-enriched cells from Nes-GFP[+] cells (mean ± SEM,

*Figure 7. Continued on next page*

*Figure 7. Continued*

*n* = 41). (**B**) Representative image of a putative HSC (asterisk) near a Nes-GFP+ cell. (**C–F**) Depletion of nestin+ cells compromises developmental HSC migration to bone marrow. (**C** and **D**) Long-term culture-initiating cell (LTCIC) assay from nestin-depleted fetal liver and bone-marrow cells. *Nes-CreER^{T2};iDTR* (red dots) and control *iDTR* (black dots) mice were exposed to tamoxifen at E14.5 and diphtheria toxin at E15.5, and liver cells (**C**) and bone marrow cells (**D**) were isolated at E17.5 (*n* = 5-6). The percentage of culture dishes that failed to generate hematopoietic colony-forming units in culture (CFU-C) is plotted against five serial dilutions of (**C**) fetal liver Lin^− Sca-1^+ cells and (**D**) nucleated bone marrow cells. HSC frequencies and p values are indicated (Pearson's chi-squared test). (**E**) Frequency of Lin^− Sca-1^+ E17.5 liver cells in mice in (**C**). (**F**) Bone marrow CFU-C content in 1-week old *Nes-CreER^{T2};R26-DTA* and control littermates treated with tamoxifen at birth (*n* = 3–7). (**G**) Expression of core HSC maintenance genes increases in perinatal Nes-GFP+ BMSCs. qPCR analysis of *Cxcl12*, stem cell factor/kit ligand (*Kitl*), angiopoietin-1 (*Angpt1*), and vascular cell adhesion molecule-1 (*Vcam1*) mRNA in CD45^- CD31^- Ter119^- GFP^{+/−} cells isolated from E18.5 and P7 *Nes-Gfp* bone marrow. (**H**) Relative *Cxcl12* mRNA expression levels in endothelial cells and Nes-GFP^{+/−} BMSCs isolated from 1-week old mice (qPCR; *n* = 2). (**I** and **J**) Relative enrichment of *Cxcl12* (**I**) and *Nestin* (**J**) mRNA expression in populations sorted from the bone marrow of P7 *Nes-Gfp;Wnt1-Cre2;R26-Tomato* compound transgenic mice. (**K**) Representative confocal image of a bone marrow section from a 1-week old (P7) *Nes-Gfp;Nes-CreER^{T2};R26-Tomato* mouse treated with tamoxifen at birth. Both sinusoidal (asterisk) and arteriolar GFP+ cells express the *Nes-CreER^{T2}-derived* Tomato (red) reporter (yellow in overlaid picture, **K'**). (**L** and **M**) Efficiency of perinatal *Cxcl12* excision by the *Nes-CreER^{T2}* driver in CD45^-Ter119^−CD31^- cells (L) and endothelial (M) cells isolated from P7 bone marrow; qPCR in CD45^-Ter119^−CD31^- cells isolated from *Cxcl12^{f/f};Nes-CreER^{T2}* (**E**) and control (**C**) littermates treated with tamoxifen at birth (*n* = 2-3). (**N** and **O**) Bone marrow CFU-C (N) and long-term HSC (O) content in P7 *Cxcl12^{f/f};Nes-CreER^{T2}* and control littermates treated with tamoxifen at birth. (**O**) Lethally-irradiated mice (CD45.1) were transplanted with 1 ×10^6 bone marrow cells from P7 *Cxcl12^{f/f};Nes-CreER* or *Cxcl12^{f/f}* mice (CD45.2), together with 1 × 10^6 recipient bone marrow cells (CD45.1). Peripheral donor-derived blood chimerism after 16 weeks is shown (*n* = 4 per group). (**E**, **L**) Each dot represents an individual mouse. (**F**, **H**–**J**) Mean ± SD. *p < 0.05, unpaired two-tailed *t* test.

The following figure supplement is available for figure 7:

**Figure supplement 1**. Neural crest-derived cells direct developmental HSC migration to the bone marrow.

most of their steady-state MSC activity after the second postnatal week. Future studies will determine the potential contribution of this MSC population to adult skeletal turnover.

Very recent data showed that *Osterix-Cre*-labeled cells in neonatal bone give rise to Nes-GFP+ MSCs (*Mizoguchi et al., 2014*; *Ono et al., 2014*); however, the *Osterix-Cre* lines used in these studies have been shown previously to mark not only osteolineage cells but also a large variety of non-osteolineage cells, including adventitial reticular cells, vascular smooth muscle cells, adipocytes, and perineural cells (*Liu et al., 2013*). It therefore remains possible that some Nes-GFP+ cells, which highly express *Osterix* mRNA (but not the protein, as we show here), also express the *Osterix-Cre* transgene; however, this might not necessarily reflect a lineage relationship or osteoblastic commitment. Moreover, it is unclear whether this *Osterix-Cre*-traced population is uniformly mesodermal or neural crest-derived. We did not find Nes-GFP+ cells within cartilage, but some GFP+ cells were associated with the innermost part of the perichondrium, a region that contains MSCs (*Maes et al., 2010*; *Yang et al., 2013*; *Zaidi and Mendez-Ferrer, 2013*). Moreover, we found perichondrial *Wnt1-Cre2*-traced cells and neural crest-derived chondrocytes in the most superficial layers of articular cartilage, suggesting that the neural crest also contributes to these mesenchymal cells outside the marrow. Neural crest-derived skeletal embryonic precursors might be Pdgfrα+ Nes-GFP^- BMSCs, since we have shown that most neural crest cells traced by *Wnt1-Cre2* are Pdgfrα+ Nes-GFP+ cells, which do not seem to contribute to fetal osteochondral lineages.

Previous studies suggested that only endochondral cells can form a hematopoietic microenvironment when implanted beneath the kidney capsule (*Chan et al., 2008*, *2013*), although bones formed by intramembranous ossification, such as the skull, can also form hematopoietic marrow without progressing through a cartilage intermediate. Notably, two surface markers used to isolate osteochondrogenic precursors in these studies, CD105 and CD90/Thy1, are particularly enriched in bone marrow Nes-GFP+ cells. Our genetic studies clearly show that neural crest-derived cells that are not yet committed to the Schwann cell lineage migrate to bone in association with developing nerve fibers and give rise to bone marrow nestin+ MSCs with specialized HSC niche functions. Constitutive deletion of the neuregulin-1 receptor ErbB3, which does not directly affect the peripheral nerves but impairs perineural migration of neural crest-derived cells (*Riethmacher et al., 1997*), reduces BMSC number and impairs developmental HSC migration from liver to bone marrow. In contrast, conditional deletion of *ErbB3* in committed glial precursors severely reduced Schwann cell numbers (*Sheean et al., 2014*) but did not affect bone marrow HSCs in our study. HSC maintenance genes are highly enriched and progressively upregulated in Nes-GFP+ Pdgfrα+ MSCs, coincident with HSC bone marrow colonization (*Christensen et al., 2004*). Perinatal deletion of nestin+ cells or blockade of their Cxcl12 production prevented HSC bone marrow seeding. Therefore, we are confident that neural crest-derived nestin+

MSCs are indeed required for developmental HSC migration and formation of the bone marrow HSC niche.

Uncertainties remain about the possible overlap among different adult bone marrow mesenchymal cells with proposed HSC niche functions, which might also differ in fetal and adult bone marrow. Two populations of adult bone marrow Nes-GFP+ cells can be separated by fluorescence intensity by microscopy. Nes-GFP^bright peri-arteriolar cells, unlike Nes-GFP^dim peri-sinusoidal cells, were found highly enriched in CFU-F activity and expression of HSC maintenance genes (*Kunisaki et al., 2013*). However, these features have been previously attributed to peri-sinusoidal cells (*Kiel et al., 2005*; *Sacchetti et al., 2007*), which, unlike arteriolar cells, express both Nes-GFP+ and leptin receptor (*Ding et al., 2012*). Alternatively, an intermediate type of vessel that connects arterioles with sinusoids near the bone surface might be highly enriched in nestin+ MSCs. These vessels, different from arterioles and sinusoids, contain CD31^hi Endomucin^hi endothelial cells and have been associated with osteoprogenitor activity (*Kusumbe et al., 2014*). We initially reported that adult bone marrow Nes-GFP+ cells were CD31− after using an intensive enzymatic digestion protocol to isolate the cells (*Mendez-Ferrer et al., 2010*). In the present study, we have used milder conditions that better preserve antigen expression. In these conditions, we detect Nes-GFP+ CD31+ putative bone marrow endothelial cells progressively increasing in number with age, as reported by others (*Ono et al., 2014*). We also find that multiple layers of Nes-GFP+ endothelial and perivascular cells likely make the arteriolar GFP signal appear brighter for GFP under the microscope. This raises the possibility that arteriolar Nes-GFP+ cells (*Kunisaki et al., 2013*) might not completely coincide with the brightest cells isolated by FACS. However, we also found that *Cxcl12* is more abundantly produced by Nes-GFP+ BMSCs than by endothelial cells, and that *Cxcl12* deletion in *Nes-CreER^T2* mice impairs developmental HSC migration by preferentially targeting BMSCs.

Different bone marrow cells have been proposed as the main producers of Cxcl12 for HSCs. One report has argued that the only relevant source of Cxcl12 for adult HSC maintenance is nestin−leptin receptor−mesenchymal progenitors targeted by the *Prx1-Cre* driver (*Greenbaum et al., 2013*). This conclusion is based on the stronger HSC depletion when *Cxcl12* was deleted using the *Prx1-Cre* line—which targets somatic lateral plate mesoderm and its derivatives, including chondrogenic and osteogenic lineages—compared to deletion in more specific stromal populations. Also, the HSC defect was specifically attributed to CD45- Lin− Pdgfrα+ Sca-1+ *Prx1-cre*-tdtomato^hi cells which were not enriched in the expression of nestin or leptin receptor but also showed no marked enrichment in the expression of *Cxcl12*, *Kitl* or other mesenchymal markers. An alternate model is that endothelial cells are needed for Cxcl12-mediated adult HSC maintenance (*Ding et al., 2012*; *Ding and Morrison, 2013*). Nonetheless, the *Tek-Cre* system used would also target endothelial cells in fetal hematopoietic organs, so some of the effects might have been exported to the bone marrow during development. The study, in essence, proposed that nestin-negative leptin receptor (*Lepr*)-*Cre*-traced mesenchymal progenitors are another key source of Cxcl12 for HSC maintenance. These studies, however, analyzed the adult bone marrow, whereas the focus of our study has been the fetal and perinatal period. Differences may also have arisen from distinct experimental settings, Cre drivers (constitutive/inducible), Cre induction regimes and reporters used. It is also likely that constitutive *Prx1-Cre* and *Lepr-Cre* lines target multiple mesenchymal derivatives and that combined deletion of *Cxcl12* in these populations would have a more pronounced effect than deletion in specific cell types; however, the responsible cell populations might not be clear yet. Conversely, the lower excision in *Nes-Cre* mice and inefficient bone marrow recombination in adult *Nes-CreER^T2* mice might not target all MSCs, or might result in compensatory actions by other Cxcl12-producing cells. In the present study, recombination efficiency in Nes-GFP+ MSCs was higher when tamoxifen was administered in neonatal *Nes-CreER^T2* mice than in adults. It has also been proposed that *Nes-Gfp* and *Nes-CreER^T2* lines might target different populations (*Ding et al., 2012*). To directly address this, we generated *Nes-Gfp*;*Nes-CreER^T2*;*R26-Tomato* triple transgenic mice that demonstrate consistent labeling using a different induction protocol. Bone marrow endothelial *Cxcl12* expression levels are significantly lower in this model, and seem unaffected by *Nes-CreER^T2*-driven excision, contrasting with the reduction in BMSCs, which was associated with decreased HSPC numbers in perinatal bone marrow. Our results thus clearly show that Cxcl12 produced by nestin+ MSCs is required for developmental HSC migration to bone marrow.

We recently reported that sympathetic neuropathy of the HSC niche is required for the manifestation of myeloproliferative neoplasms, disorders previously considered to be autonomously driven by mutated

HSCs and typically associated with excessive fibroblasts and osteoblasts in the bone marrow. During this pathogenesis, bone marrow nestin$^+$ cells do not seem to differentiate into fibroblasts or osteoblasts, but instead activate the Schwann cell program as a consequence of the neuroglial damage caused in the bone marrow by mutated HSCs (*Arranz et al., 2014*). These changes could be explained by a neural crest contribution found for HSC niche-forming MSCs and suggest the possible re-programming of these cells towards the closest ontogenically-related linages during the pathogenesis of these disorders.

In summary, this study designates separate biologic functions to ontogenically distinct populations of MSCs, and demonstrates that not all MSCs are alike. In the appendicular skeleton, nestin$^-$ MSCs derived from the mesoderm have a primarily osteochondroprogenitor function. In contrast, a distinct population of neural crest-derived nestin$^+$ MSCs contributes to directed HSC migration through the secretion of the chemokine Cxcl12 to ultimately establish the HSC niche in the neonatal bone marrow. These niche-forming MSCs share a common origin with sympathetic neurons and Schwann cells, an ontogenic relationship that underscores our earlier observations on the sympathetic control of HSC niche function (*Mendez-Ferrer et al., 2008*, *2010*; *Arranz et al., 2014*). Future studies will also determine whether tight regulation of other peripheral adult stem cell niches by the nervous system also builds upon an ontogenic relationship of their components.

## Materials and methods

### Animals

Mouse lines used in this study (please see *Supplementary file 1* for a detailed list of mouse strains used in this study) included *Nes-Gfp* (*Mignone et al., 2004*), *Nes-CreER$^{T2}$* (*Balordi and Fishell, 2007*), *Sox10-CreER$^{T2}$* (*Matsuoka et al., 2005*), *Col2.3-Cre* (*Dacquin et al., 2002*), *Dhh-Cre* (*Jaegle et al., 2003*), *RCE-loxP* (*Sousa et al., 2009*), *LSL-KFP* (*Dieguez-Hurtado et al., 2011*), *R26-DTA* (*Brockschnieder et al., 2006*), *Cxcl12$^{floxed}$* (*Tzeng et al., 2010*), *Erbb3$^{floxed}$* (*Sheean et al., 2014*), *Erbb3*-null (*Riethmacher et al., 1997*), and *129S4.Cg-Tg(Wnt1-cre)2Sor/J, C57BL/6-Gt(ROSA)26Sor$^{tm1(HBEGF)Awai}$/J, B6.Cg-Gt(ROSA)26Sor$^{tm14(CAG-tdTomato)Hze}$/J,* wild-type CD1 and wild-type C57BL/6J (Jackson Laboratories). Material and methods were approved by the Animal Care and Use Committees of the Spanish National Cardiovascular Research Center and Comunidad Autónoma de Madrid (PA-47/11 and ES280790000176).

### Embryo analysis and genetic inducible fate mapping

Embryos were dissected as previously described (*Isern et al., 2008*). Briefly, selected intercrosses between mice carrying the alleles of interest were set and the morning of detection of the vaginal plug was considered as day 0.5 of gestation. We preferentially used paternal transgene transmission, by mating compound or simple transgenic males with females of wild-type background (C57BL/6 or CD1). Inducible lineage tracing studies were conducted as follows. Tamoxifen (T5648; Sigma, St. Louis, MO) was dissolved in corn oil at a final concentration of 20 mg/mL and given to pregnant dams by oral gavage (100-150 mg/kg) on the morning of the indicated stages. For neonatal induction, mothers of newborn pups were given tamoxifen (by oral gavage, 4 mg) on days 1 and 3 after delivery.

### Histology

Dissected tissues for histology were fixed in 2% paraformaldehyde at 4°C, cryopreserved by consecutive equilibration in 15% and 30% sucrose, and snap frozen and embedded in OCT compound (Tissue-Tek). In some cases, fixed frozen limbs or sterna were trimmed sequentially from both sides to expose the central medullar cavity and processed further for whole-mount fluorescence staining. Cryostat sections (15 µm) were prepared and processed for immunostaining or regular hematoxylin–eosin staining. Oil red O staining was performed as described (*Isern et al., 2013*).

### Immunohistochemistry

Cryostat sections were stained using standard procedures. Briefly, tissues were permeabilized for 5-10 min at room temperature (RT) with 0.1% Triton X-100 and blocked for 1 h at RT with TNB buffer (0.1 M Tris–HCl, pH 7.5, 0.15 M NaCl, 0.5% blocking reagent, Perkin Elmer, Waltham, MA). Primary antibody incubations were conducted either for 1-2 h at RT or overnight (o/n) at 4°C. Secondary antibody incubations were conducted for 1 h at RT. Repetitive washes were performed with PBS + 0.05% Tween-20. Stained tissue sections were counterstained for 5 min with 5 µM DAPI and rinsed with PBS. Slides were mounted in Vectashield Hardset mounting medium (Vector Labs, Burlingame, CA) and sealed with nail polish.

For whole-mount staining of thick-sectioned tissue pieces, all the incubations, including permeabilization and blocking, were performed o/n at 4°C with gently agitation, and washing steps were extended. Specimens were mounted in glass bottom dishes (Mat-Tek, Ashland, MA).

## Immunofluorescence

Fetal bone marrow and fetal liver sections were stained following standard procedures. Antibodies used are indicated in the table below. SLAM staining was performed in bone marrow sections from neonate mice. Slides were first blocked in 20% goat serum in PBS for 45 min. Endogenous avidin and biotin were blocked with an Avidin/Biotin Blocking Kit (Vector Labs) for 30 min with each reagent, washing 3 times with PBS in between. Slides were then incubated with rat anti-mouse CD150 antibody (Biolegend, San Diego, CA) at 1:50 dilution in goat blocking buffer for 2 h, and after washes with goat anti-rat IgG conjugated to Alexa555 (Molecular Probes, Eugene, OR) at 1:200 dilution in 20% goat serum in PBS for 1 h. Slides were then blocked in 20% rat serum in PBS for 10 min and incubated for 1 h with hamster anti-mouse biotin-conjugated CD48 (Abcam, UK) and the Biotin Mouse Lineage Panel (BD Pharmingen, San Jose, CA), which includes rat anti-mouse B220, rat anti-mouse CD3, rat anti-mouse Gr1, rat anti-mouse Mac-1, and rat anti-mouse Ter119 antibodies, each at 1:200 dilution in rat blocking buffer. Cy5-conjugated streptavidin (Molecular Probes) was added at 1:200 in rat blocking buffer for 30 min. Finally, slides were incubated with DAPI (1:1000 dilution of 5 mg/ml stock) for 10 min at room temperature and mounted in Vectashield Mounting Medium (Vector Labs). Antibodies used for Immunohistochemistry:

| Name | Type | Company |
|---|---|---|
| TH | Rabbit pAb | Millipore |
| GFAP | Rabbit pAb | Dako |
| CD31 | Rat mAb | BD Pharmingen |
| S100 | Rabbit pAb | Dako |
| Collagen type IV | Rabbit pAb | Millipore |
| Ki67 | Rabbit pAb | Abcam |
| Anti-KFP | Rabbit pAb | Evrogen |
| CD150 | Rat mAb | Biolegend |
| CD48 | ArHm mAb | Abcam |
| Lineage-biotin | Rat mAb | BD Biosciences |
| Tuj1 | Mouse mAb | Promega |
| Anti-GFP | Rabbit pAb | Abcam |
| c-kit | goat pAb | R&D |
| Nestin | Rabbit pAb | Abcam |
| α-SMA | Mouse mAb | Sigma |

## Imaging

Confocal images of fluorescent staining were acquired with a laser scanning confocal microscope (Zeiss LSM 700, 10×/0.45, 25×/0.85) or with a multi-photon Zeiss LSM 780 microscope (10×/0.7, 20×/1.0). Optical z-stack projections were generated with the Zen2011 software package (Zeiss, Germany) using a maximal intensity algorithm. Wide-field views of whole-mount specimens were imaged with a Leica MZFLIII stereomicroscope equipped with an Olympus DP71 color camera. Images were post-processed and quantified using ImageJ (*Schneider et al., 2012*) and Photoshop (Adobe, San Jose, CA).

## Preparation of fetal and neonatal bone marrow cell suspensions

Fetal skeletal elements were sub-dissected from fetuses, homogenized by cutting, and digested for 15-30 min at 37°C with shaking in 0.25% collagenase (StemCell Technologies, Canada). Postnatal bone specimens were cleaned from surrounding tissue, crushed in a mortar with a pestle, and collagenase-digested for 45-60 min at 37°C, with constant agitation. After enzymatic treatment, skeletal preparations were filtered through a 40-μm cell strainer and undigested bone material was discarded. The resulting bone marrow-enriched cell suspensions were pelleted, washed twice, and resuspended in FACS staining buffer (2% FCS in PBS) for further analysis.

## Flow cytometry

Dispersed bone marrow cell preparations were stained in FACS buffer for 15-30 min on ice with selected multicolor antibody cocktails (see below), washed, and resuspended with streptavidin conjugates when necessary. Stained cells were pelleted and resuspended in buffer containing DAPI to exclude dead cells. Cell cycle was analyzed by first isolating defined stromal populations by FACS, and then acquiring the cell cycle profile after staining the sorted populations with Hoechst 33342. Flow cytometry analysis and FACS were done in FACS CantoII or LSRFortessa machines (BD Biosciences) equipped with Diva Software (BD Biosciences) or in a FACS AriaII cell sorter (BD Biosciences). Data were analyzed using Diva and FlowJo (Tree Star, Inc, Eugene, OR). Antibodies used for cytometry:

| Name | Clone | Company |
| --- | --- | --- |
| CD45-APC/Cy7 | 104 | BD Biosciences |
| CD45-APC | 104 | BD Biosciences |
| CD31-APC | MEC 13.3 | BD Biosciences |
| Ter119-APC | Ter119 | BD Biosciences |
| CD140a-biotin | APA5 | eBioscience |
| CD140a-APC | APA5 | Biolegend |
| CD90.2-APC | 53-2.1 | eBioscience |
| Ly6a-PE | E13–161.7 | BD Biosciences |
| Vcam1-PE | 429 (MVCAM.A) | Biolegend |
| Streptavidin-PE | -- | BD Biosciences |
| Lineage cocktail-biotin | | BD Biosciences |

## CFU-F and CFU-OB assays

For fibroblast colony-forming unit (CFU-F) assays, bone marrow cell suspensions were FACS sorted directly into 6-well plates at a cell density of 100–500 cells/cm$^2$ and cultured in maintenance medium (α-MEM/15% FCS with antibiotics). After 10-12 days in culture, adherent cells were fixed with 100% methanol and stained with Giemsa stain (Sigma) to reveal fibroblast clusters. Colonies with more than 50 cells were scored as CFU-Fs. For osteoblast colony-forming unit (CFU-OB) assays, plated cells were cultured in maintenance medium in the presence of 1 mM L-ascorbate-2-phosphate. All cultures were maintained with 5% CO$_2$ in a water-jacketed incubator at 37°C, and medium was changed weekly. After 25 days in culture, cells were fixed and stained with alizarin red or alkaline phosphatase (*Isern et al., 2013*).

## Hematopoietic progenitor assays

Single cell suspensions were prepared from bone marrow and mixed with methylcellulose-containing medium with cytokines (*Casanova-Acebes et al., 2013*). Cells (5-7.5 × 10$^4$) were plated in duplicate 35 mm dishes (Falcon, BD) and incubated under 20% O$_2$ and 5% CO$_2$ in a water-jacketed incubator. Hematopoietic colonies (CFU-Cs) were scored after 6-7 days in culture.

## Long-term culture-initiating cell assay

Long-term culture-initiating cell assay was performed as described (*Woehrer et al., 2013*). Briefly, the feeder fetal stromal cell line AFT024 (kindly provided by Dr. K. Moore) was maintained as previously described (*Nolta et al., 2002*). One week before use, the feeders were irradiated (15 Gy) with a $^{137}$Cs irradiator and seeded in 96-well plates at confluency. After 7-10 days, five serial dilutions (each with 16 replicates) of sorted fetal liver Lin$^-$ Sca1$^+$ cells and bone marrow nucleated cells were seeded on the irradiated feeders and cultured with Myelocult M5300 supplemented with 10$^{-6}$ M hydrocortisone (StemCell Technologies) and 1% penicillin-streptomycin (Invitrogen, Carlsbad, CA). Cultures were maintained for four weeks at 33°C under 20% O$_2$ and 5% CO$_2$ in a water-jacketed incubator. Medium was half-changed weekly. Each well was then trypsinized for 10 min, washed with PBS, and plated for the hematopoietic progenitor assay. Twelve days after plating, the percentage of culture dishes in each experimental group that failed to generate CFU-Cs was plotted against the number of test cells. The frequencies of long-term culture-initiating cells were calculated by the Newton–Raphson method of maximum likelihood and Poisson statistics (using L-CalcTM software; StemCell Technologies) as the reciprocal of the number of test wells that yielded a 37% negative response.

## Cell culture and in vitro differentiation

Primary bone marrow cells were obtained from dissected bones using a mortar. All cultures were maintained at 37°C with 20% $O_2$, 5% $CO_2$ in a water-jacketed incubator.

To obtain CFU-Fs and CFU-OBs, $0.5 \times 10^6$ bone marrow nucleated cells were seeded in each well of a 12-well plate with α-MEM supplemented with 1% penicillin-streptomycin, 15% FBS (Invitrogen), and 1 mM L-ascorbic acid 2-phosphate (Sigma). Half medium was replaced every 5 days. The numbers of CFU-Fs and CFU-OBs were scored after 10 and 28 days in culture, respectively.

CFU-F cultures were fixed in methanol for 10 min at room temperature. Cultures were stained with Giemsa diluted 1:10 in phosphate buffer, pH 6.8, for 10 min at 37°C. CFU-F colonies (those with more than 50 cells) were counted the next day.

CFU-OB cultures were fixed with 4% paraformaldehyde (PFA) for 5 min at room temperature. von Kossa staining was performed by adding 5% $AgNO_3$ to the culture and exposing plates to UV radiation for 20 min. Cells were then incubated with 5% $(NH_4)_2S_2O_3$ in distilled water for 5 min and counterstained with 2% eosin. For Alizarin Red staining, cells were incubated with 2% alizarin red reagent (Sigma) in distilled $H_2O$ for 15 min. For alkaline phosphatase staining, Sigma Fast BCIP/NBT substrate (Sigma) was added to cell cultures followed by incubation in the dark for 15 min.

## Nucleic acid purification and qPCR

RNA from CFU-Fs and osteoblast cultures was extracted with Trizol reagent (Sigma-Aldrich) and purified on RNeasy mini columns (Qiagen, Netherlands). An on-column DNase digest (Qiagen) was performed before the clean-up step to eliminate residual genomic DNA. For osteoblast cultures, mRNA was extracted with the Dynabead mRNA DIRECT kit (Invitrogen). cDNA was generated using High Capacity cDNA Reverse Transcription reagents (Applied Biosystems, Waltham, MA). qPCR was performed in triplicate with SYBRgreen Universal PCR Master Mix (Applied Biosystems), using primers optimized for each target gene. The expression level of each gene was determined by using the relative standard curve method. Briefly, a standard curve was performed by doing serial dilutions of a mouse reference total RNA (Clontech, Palo Alto, CA). The expression level of each gene was calculated by interpolation from the standard curve. Relative quantifications of each transcript were obtained by normalizing against *Gapdh* transcript abundance, using the standard curve method. The sequences of oligonucleotides for qPCR are detailed below.

| Target gene | Symbol | Forward | Reverse |
|---|---|---|---|
| *Alkal.Phosphat.* | Alpl | CACAATATCAAGGATATCGACGTGA | ACATCAGTTCTGTTCTTCGGGTACA |
| *Osterix* | Sp7 | ATGGCGTCCTCTCTGCTTGA | GAAGGGTGGGTAGTCATTTG |
| *Runx2* | Runx2 | TTACCTACACCCCGCCAGTC | TGCTGGTCTGGAAGGGTCC |
| *Rank ligand* | Rankl | CAGCATCGCTCTGTTCCTGTA | CTGCGTTTTCATGGAGTCTCA |
| *Gpnmb* | Gpnmb | CCCCAAGCACAGACTTTGAG | GCTTTCTGCATCTCCAGCCT |
| *Osteocalcin* | Bglap | GGGCAATAAGGTAGTGAACAG | GCAGCACAGGTCCTAAATAGT |
| *Osteoglycin* | Ogn | ACCATAACGACCTGGAATCTGT | AACGAGTGTCATTAGCCTTGC |
| *Rank* | Rank | TGCAGCTCAACAAGGATACG | GAGCTGCAGACCACATCTGA |
| *TRAP* | Acp5 | CAGCAGCCAAGGAGGACTAC | ACATAGCCCACACCGTTCTC |
| *Cathepsin k* | Ctsk | GGCCTCTCTTGGCCATA | CCTTCCCACTCTGGGTAG |
| *Mmp-9* | Mmp9 | CGTCGTGATCCCCACTTACT | AACACACAGGGTTTGCCTTC |
| *Ppar gamma* | Pparg | ACCACTCGCATTCCTTTGAC | TGGGTCAGCTCTTGTGAATG |
| *Adiponectin* | Adipoq | TGTTCCTCTTAATCCTGCCCA | CCAACCTGCACAAGTTCCCTT |
| *Adipsin* | Cfd | TGCATCAACTCAGAGTGTCAATCA | TGCGCAGATTGCAGGTTGT |
| *Sox9* | Sox9 | GAACAGACTCACATCTCT | GTGGCAAGTATTGGTCAA |
| *Col2a1* | Col2a1 | GTGGAGCAGCAAGAGCAAGGA | CTTGCCCCACTTACCAGTGTG |
| *Aggrecan* | Acan | CACGCTACACCCTGGACTTTG | CCATCTCCTCAGCGAAGCAGT |
| *Cxcl12* | Cxcl12 | CGCCAAGGTCGTCGCCG | TTGGCTCTGGCGATGTGGC |
| *Kit ligand* | Kitl | CCCTGAAGACTCGGGCCTA | CAATTACAAGCGAAATGAGAGCC |
| *Angiopoietin 1* | Angpt1 | CTCGTCAGACATTCATCATCCAG | CACCTTCTTTAGTGCAAAGGCT |
| *Nestin* | Nes | GCTGGAACAGAGATTGGAAGG | CCAGGATCTGAGCGATCTGAC |

## Primary sphere-forming cultures

For sphere formation, cells were plated at clonal density (<1000 cells/cm$^2$) in ultra-low adherent 35 mm dishes (StemCell Technologies). The growth medium consisted of DMEM/F12 (1:1) mixed 1:2 with human endothelial serum-free medium (Invitrogen) and contained 15% chicken embryo extract, prepared as described (*Stemple and Anderson, 1992*; *Pajtler et al., 2010*); 0.1 mM ß-mercaptoethanol; 1% non-essential aminoacids (Sigma); 1% N2 and 2% B27 supplements (Invitrogen); recombinant human fibroblast growth factor (FGF)-basic; recombinant human epidermal growth factor (EGF); recombinant human platelet-derived growth factor (PDGF-AB); recombinant human oncostatin M (227 a.a. OSM) (20 ng/ml); and recombinant human insulin-like growth factor-1 (IGF-1; 40 ng/ml) (Peprotech, Rocky Hill, NJ). The cultures were kept at 37°C under 5% $CO_2$, 20% $O_2$ in a water-jacketed incubator, and were left untouched for one week to prevent cell aggregation in low density cultures. Medium was half-changed weekly. Mesenspheres were scored on days 10–14.

## In vitro differentiation of Schwann cells from bone marrow precursors

We used an adaptation of the original method (*Biernaskie et al. 2006*). Defined stromal populations were isolated based on GFP and Pdgfrα expression from collagenase-treated bone marrow of *Nes-Gfp* neonates. Sorted cells were plated onto laminin/polylysine-coated chamber slide dishes (Labtek) and allowed to attach and expand in SKP medium I. After 3 days, cells were changed to SKP medium II (containing neuregulin-1 at 50 ng/mL) and allowed to differentiate further for >10 days. In vitro-generated Schwann cells were defined by morphology as thin and elongated cells. After differentiation, cells were fixed in 4% PFA, gently permeabilized with Triton X-100, and stained for immunofluorescence with anti-glial fibrillary acidic protein (Gfap) antibody (Dako, Carpinteria, CA).

## In vitro differentiation of bone marrow mesenchymal cells

Defined stromal populations were isolated based on GFP and Pdgfrα expression from collagenase-treated bone marrow of *Nes-Gfp* neonates and plated directly onto plastic dishes to allow attachment of fibroblasts. Adherent cells were cultured for 7-14 days in regular α-MEM supplemented with 15% FBS. In some cases, recombinant human PDGF was added at 20 ng/mL. At the end of the culture period, cells were fixed and stained with Oil red O to reveal adipocytes and counterstained with hematoxylin.

## RNA-Seq

For next-generation sequencing, total RNA was isolated using the Arcturus Picopure RNA isolation kit (Life Technologies, Carlsbad, CA) from small numbers of sorted cells (15,000-80,000), obtained from neonatal *Nes-Gfp* bone marrow preparations (two biological replicates). Each independent set of samples was obtained from pooled skeletal elements (long bones and sterna) from multiple littermates.

### RNA-Seq library production

Amplified cDNA was prepared with the Ovation RNA-Seq System V2 Kit (NuGEN Technologies Inc., San Carlos, CA) followed by sonication and TrueSeq DNA Library Preparation Kit from Illumina. The quality, quantity, and the size distribution of the Illumina libraries were determined using the DNA-1000 Kit (Agilent Bioanalyzer). Libraries were sequenced on a Genome Analyzer IIx (Illumina Inc., San Diego, CA) following the standard RNA sequencing protocol with the TruSeq SBS Kit v5. Fastq files containing reads for each library were extracted and demultiplexed using the Casava v1.8.2 pipeline.

### RNA-Seq analysis

Sequencing adaptor contaminations were removed from reads using the cutadapt software tool (MIT), and the resulting reads were mapped and quantified on the transcriptome (NCBIM37 Ensembl genebuild 65) using RSEM v1.17 (*Li and Dewey, 2011*). Only genes with >2 counts per million in ≥2 samples were considered for statistical analysis. Data were then normalized and differential expression assessed using the function Voom from the bioconductor package Limma (*Law et al., 2014*; *Smyth, 2005*). We considered those genes as differentially expressed with a Benjamini–Hochberg adjusted p-value ≤0.05.

### Principal component analysis (PCA) comparison with previously published data

Normalized RNA-Seq data were compared via principal component analysis (PCA) with previously published array expression data (*Table 1*). GEO data sets were downloaded and pre-processed using the GEOquery Bioconductor package (*Davis and Meltzer, 2007*). Data sets were adjusted to the same intensity range, as previously described (*Heider and Alt, 2013*). Batch correction was applied to the

data sets using ComBat (*Johnson et al., 2007*) on the log2-normalized GEO data sets together with the log2-normalized counts from each RNA-Seq experiment.

## Acknowledgements

The authors express sincere thanks to the following investigators for their generous donation of mice: G.E. Enikolopov (*Nes-Gfp*), G. Fishell (*Nes-CreER^T2^*, *RCE-loxP*), S. Ortega (*LSL-KFP*), D. Riethmacher (*iDTA*), V. Pachnis (*Sox10-CreER^T2^*), S. MacKem (*Hoxb6-CreER^T2^*), T. Müller and C. Birchmeier (*Erbb3^−/−^*), D.N. Meijer (*Dhh-Cre*), S. Rocha and A. García-Arroyo (*R26-Tomato*). We are grateful to S. González-Hernández, O. Pérez-Howell, J.M. Ligos, A.B. Ricote, and the CNIC Genomics Unit for technical assistance, to members of SMF lab for helpful discussions, to M. Zaidi, J.B. Aquino, I. Adameyko, P. Ernfors, M. Torres, I. Delgado, L. Carramolino, and M. García-Fernández for helpful advice and support, and S. Bartlett for editing the manuscript. This work was supported by the Spanish Ministry of Economy and Competitiveness through the Fundación Centro Nacional de Investigaciones Cardiovasculares Carlos III, SB2010-0023 grant to J.I, Plan Nacional grants BFU2012-35892 to J.I. and SAF-2011-30308 to S.M.-F., Ramón y Cajal Program grants RYC-2011-09209 to J.I. and RYC-2009-04703 to S.M.-F. The Marie Curie Career Integration Program (FP7-PEOPLE-2011-RG-294262), ConSEPOC-Comunidad de Madrid grant S2010/BMD-2542 to S.M.-F., and the Spanish Network of Cell Therapy (TerCel). A.G.-G. received a fellowship from Fundación Ramón Areces and is currently supported by Fundación La Caixa. S.M.-F. is supported in part by an International Early Career Scientist grant from the Howard Hughes Medical Institute.

## Additional information

### Funding

| Funder | Grant reference number | Author |
| --- | --- | --- |
| Howard Hughes Medical Institute | 55007426 | Simón Méndez-Ferrer |
| Ministerio de Economía y Competitividad | SAF-2011-30308 | Simón Méndez-Ferrer |
| Ministerio de Economía y Competitividad | BFU2012-35892 | Joan Isern |
| Ramon y Cajal Program | RYC-2011-09209 | Joan Isern |
| Ramon y Cajal Program | RYC-2009-04703 | Simón Méndez-Ferrer |
| Marie Curie Actions | FP7-PEOPLE-2011-RG-294262 | Simón Méndez-Ferrer |
| ConsEPOC-Comunidad de Madrid | S2010/BMD-2542 | Simón Méndez-Ferrer |
| Fundación Ramón Areces | | Andrés García-García |
| Fundación La Caixa | | Andrés García-García |
| Ministerio de Educatión | SB2010-0023 | Joan Isern |
| Spanish Cell Therapy Network | TerCel | Simón Méndez-Ferrer |

The funders had no role in study design, data collection and interpretation, or the decision to submit the work for publication.

### Author contributions

JI, Designed and performed most experiments, analyzed data and wrote the manuscript, Conception and design, Acquisition of data, Analysis and interpretation of data, Drafting or revising the article; AG-G, AMM, LA, DM-P, CT, FS-C, Performed experiments, Acquisition of data; SM-F, Conceived the overall study. Contributed most reagents, designed experiments, analyzed data and wrote the manuscript, Conception and design, Acquisition of data, Analysis and interpretation of data, Drafting or revising the article, Contributed unpublished essential data or reagents

### Author ORCIDs

Simón Méndez-Ferrer, ⓘD http://orcid.org/0000-0002-9805-9988

## Ethics

Animal experimentation: Experimental procedures were approved by the Animal Care and Use Committees of the Spanish National Cardiovascular Research Center and Comunidad Autónoma de Madrid (PA-47/11 and ES280790000176).

## Additional files

### Supplementary file

• Supplementary file 1. Summary of mouse strains used in this study.

### Major datasets

The following dataset was generated

| Author(s) | Year | Dataset title | Dataset ID and/or URL | Database, license, and accessibility information |
| --- | --- | --- | --- | --- |
| Joan Isern and Simon Mendez-Ferrer | 2014 | The neural crest is a source of mesenchymal stem cells with specialized hematopoietic stem-cell-niche function | http://www.ncbi.nlm.nih.gov/geo/query/acc.cgi?acc=GSE61695 | Publicly available at NCBI Gene Expression Omnibus. |

The following previously published datasets were used

| Author(s) | Year | Dataset title | Dataset ID and/or URL | Database, license, and accessibility information |
| --- | --- | --- | --- | --- |
| Fumio Arai and Toshio Suda | 2010 | Gene expression profile of murine bone marrow endosteal populations | http://www.ncbi.nlm.nih.gov/geo/query/acc.cgi?acc=GSE17597 | Publicly available at NCBI Gene Expression Omnibus. |
| Wislet-Gendebien S, Laudet E, Neirinckx V, Leprince P, Glejzer A, Poulet C, Hennuy B, Sommer L, Shakova O, Rogister B | 2011 | Mesenchymal stem cells and neural crest stem cells from adult bone marrow: characterization of their surprising similarities and differences | http://www.ncbi.nlm.nih.gov/geo/query/acc.cgi?acc=GSE30419 | Publicly available at NCBI Gene Expression Omnibus. |
| Makita T, Sucov HM, Gariepy CE, Yanagisawa M, Ginty DD | 2008 | Role of Endothelin in SCG axon pathfinding | http://www.ncbi.nlm.nih.gov/geo/query/acc.cgi?acc=GSE10360 | Publicly available at NCBI Gene Expression Omnibus. |
| Simon Mendez-Ferrer and Paul S Frenette | 2010 | Expression profile in bone marrow Nestin-GFP cells | http://www.ncbi.nlm.nih.gov/geo/query/acc.cgi?acc=GSE21941 | Publicly available at NCBI Gene Expression Omnibus. |
| Ding L, Morrison SJ | 2012 | Scf-GFP + cells from the bone marrow and whole bone marrow microarray | http://www.ncbi.nlm.nih.gov/geo/query/acc.cgi?acc=GSE33158 | Publicly available at NCBI Gene Expression Omnibus. |
| Buchstaller J, Sommer L, Bodmer M, Hoffmann R, Suter U, Mantei N | 2004 | NCSC-SC development | http://www.ncbi.nlm.nih.gov/geo/query/acc.cgi?acc=GSE972 | Publicly available at NCBI Gene Expression Omnibus. |

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
