## [Decision Letter]

Thank you for sending your work entitled “The neural crest is a source of mesenchymal stem cells with specialized hematopoietic stem cell niche function” for consideration at *eLife.* Your article has been favorably evaluated by Janet Rossant (Senior editor) and 3 reviewers, one of whom is a member of our Board of Reviewing Editors.

The Reviewing editor and the other reviewers discussed their comments before we reached this decision, and the Reviewing editor has assembled the following comments to help you prepare a revised submission.

While all reviewers concur on the importance of your work and the potential impact in the field, two out of three are not satisfied by the quality of the images that support your conclusions. In a lineage tracing study this is a crucial requirement since, at low magnification, as all the images are, it is very difficult if not impossible to demonstrate co-localisation of the tracer with the tissue specific antigen under observation. Confocal, high magnification analysis would be required to substantiate the results in virtually all the figures presented. Moreover, even at low magnification there are clear inconsistencies: for example expression of CD31 is not known to occur in non-endothelial cells and thus a double staining should have been performed. Also, while Osterix appears not to be expressed in lateral plate derived cells (but very few cells are shown), the co-expression with Wnt1 labeled neural crest is partial and doubtful, making the main conclusion of the whole work particularly weak.

In addition all reviewers found the manuscript difficult to read and practically inaccessible to a non-specialised audience. The reviews are included below to assist with your revisions.

*Reviewer #1*:

This manuscript reports an interesting and potentially important finding on functional heterogeneity and developmental origin of bone marrow stromal cells.

However the main and potentially ground breaking finding, i.e. that Nestin+ MSC are derived from neural crest and support hematopoiesis rather than building the bone itself, is not supported by the data presented, that are of poor quality and do not allow to draw final conclusions on the developmental origin of these cells.

Moreover, the manuscript is poorly written, difficult to follow and would greatly benefit from careful editing and, at least in some parts, re-writing.

1) Authors report that a small but increasing fraction of GFP+ also express CD31 and that this fraction increases with age. Since Nestin is not known to be expressed in endothelial cells, a co-staining with also anti-nestin antibody should be performed and co-localization investigated at confocal level, since this bears important implications on the fidelity of the GFP transgene.

2) The text describing data presented in Figure 2 is particularly confusing and almost impossible to interpret: “These mice and newborn Nes-gfp embryos were analyzed for osterix protein expression, which marks cells committed to the osteoblastic lineage. In contrast, cells genetically traced by the regulatory elements of Osterix gene do not only comprise osteoblastic cells, but also adventitial reticular cells, vascular smooth muscle cells, adipocytes and perineural cells (43). Unlike cells derived from lateral plate mesoderm, fetal limb bone marrow Nes-GFP+ cells did not express osterix protein (Figure 2) and therefore could not be considered osteoblast precursors.”

After describing the Hoxb6-CreERT mice (labelling lateral mesoderm derived cells) the Authors show no co-localization of Nes-GFP and Osterix, and then “presumed” co-localization of Hoxb6 (i.e. lateral plated) derived cells and Osterix (Figure 2). They conclude that because Nestin-GFP+ cells do not co-localize with Hoxb6 labelled cells they cannot be of lateral plate origin. This is a crucial point and particularly weak. First of all, at the magnification shown it is impossible to draw any conclusion. In A', very few GFP+ cells are visible and in an area where they are quite distant from Osx+ cells.

On the other hand, in B' only a very small minority of HoxB6 derived cells express Osx (and at that magnification it is hard to be sure): in most case the colour overlap seems partial, as it may originate from two cells one above the other. Even assuming that there is co-localization in few cells, what about all the others Osx+ cells? Even though recombination is never 100% efficient, one would conclude that the very large majority of Osx expressing cells do not derive from lateral mesoderm. Maybe they originate from paraxial mesoderm, but then another Cre would be required or the origin of Osx expressing cells remains largely unidentified in this work.

3) The overlap between Nestin-GFP and Nestin-Cre derived cells is only minor and, again, at that magnification it is impossible to draw solid conclusions. It is possible that Nestin-Cre derived cells have turned off the expression of Nestin gene, but then what have they become? Conversely, many Nestin-GFP+ do not seem to have originated from cells previously expressing Nestin. Even the example at “high” magnification (Figure 2) shows one cell (**) that is clearly double labelled and another example (*) where there are two neighbouring but clearly distinct cells. These data are not convincing at all of a reliable lineage tracing.

4) Data on Wnt1-Cre appear more convincing, but again, a high magnification of a double-labelled cell at the confocal level would be necessary, especially for osteocytes (Cartilage staining is quite convincing even at low mag). However, FACS analysis shown in Figure 4 has problems. The Wnt1-Cre/Tomato clearly shows a dim (continuous with background) and a very bright population, both gated together to show that 62% of these are Nes-GFP+. I would have liked to see whether the Nes-GFP+ cells are equally represented in the bright and the dim populations. This is critical because the results are highly dependent on the specific gating. Most importantly, the IF shown in Figure 4 is highly unconvincing and does not support the claim that Nes-GFP+ are derived from Wnt1 expressing cells. The very few Tomato+ cells do not really co-localize with GFP+ cells and the examples shown in D look like at most closely located cells. Finally, the FACS analysis for Sox10-Cre also shows that gating has included both bright cells, clearly separated from the negative peak and the shoulder of this. I would like to see whether the bright cells only are GFP+.

5) The final sections of the Results, though also suffering of the poor quality of the images, are more convincing but less original. Furthermore DTR experiments are tricky, first because killing Nestin expressing cells may have also consequences in other tissue and possibly systemic effects. In addition, and this criticism applies to all of these studies, developmental biologists seem to be unaware of the “bystander effect”, well studied by gene therapists, where killing a specific cell, often causes the death of the nearest cells, thus mudding the results.

*Reviewer #2*:

The manuscript by Isern et al describes the results of experiments examining the developmental origins (fetal, neonatal and adult) of the mesenchymal cells that are osteogenic and that support the hematopoietic system in the bone marrow. The experiments mainly utilize the nestin-gfp reporter mouse model in addition to other mouse models in which induced recombination mediated reporter expression defines specific neural crest derived populations. The manuscript is filled with information but it is sometimes difficult to understand how it is related to the main question. For example, it is not explained why CD31 immunostaining is performed on nestin gfp tissues; what cell types does it mark? Convention is the endothelial and hematopoietic cells express CD31, not MSC. The Discussion provides some information about levels of CD31 expression but this comes rather late in the manuscript.

In the Results section that describes BM nestin+ cells do not contribute to fetal endochondrogenesis, the RCE reporter is used, what is this? No definition is provided. Also, later in this paragraph, the sentence beginning with “In contrast, cells…” seems misplaced and confusing, making this text and results hard to follow. Similarly, it is not understood why a Wnt Cre line was used. What is the rationale? I seem to miss the connection. It would be helpful to reorganize and perhaps move some information from the main text to the Discussion or leave it out if it is not directly related to the question. It would be helpful to the reader if the manuscript could be more direct in describing the experimental rationale.

Altogether, the authors have done a large amount of work to show that there are different component populations of mesenchymal cells that contribute to the development of the long bones, bone marrow and the supportive hematopoietic microenvironment. This is an important set of results that supports a new conceptual model for the development of the hematopoietic supportive microenvironment.

*Reviewer #3*:

This manuscript identifies the ontological source of an important bone marrow (BM) niche cells previously identified by the senior author. These cells are the Nestin-GFP+ (Nes-GFP+) cells in the adult BM that are a source of MSC and niche support. The Mendez-Ferrer group has extended this work to the developing fetal and early postnatal bone and observed exciting and enlightening results. They demonstrate that trunk neural crest cells are the developmental source for Nes-GFP+ MSC in the BM, in addition to their known contributions to peripheral sympathetic neurons and Schwann cells. Interestingly, all three of these cells types are important BM niche components. As such they share an ontological relationship that could theoretically extend to other niche components elsewhere in the body. Nes-GFP+ cells were characterized in the developing bone (E17.5), at P0 and at P7 with a variety of lineage tracing and Cre deletion transgenics in addition to cell surface IHC and flow cytometry. From a wide array of models they show:

1) Nes-GFP+ cells do not exhibit osteochondral progenitor activity. Nes-GFP- cells that derive from lateral plate mesoderm possess this activity in fetal bone.

2) Neural crest contribution to postnatal BM-MSCs was definitively demonstrated by genetic fate mapping with Wnt1-Cre2 and Sox10-CreERT2.

3) Cell surface and mRNA-seq studies of PDGFRa+/- and Nes-GFP+/- cells revealed that there are two Nes-GFP+ neural crest derived cells in postnatal BM. Schwann cell precursors are in the PDGFRa- fraction while PDGFRa+ cells contain the MSCs.

4) Neural crest cells migrate along developing nerves to the fetal bone before commitment to the Schwann cell lineage and give rise to niche forming MSCs.

5) Cxcl12 expression was shown to be 80 and 20-fold higher in Nes-GFP+ cells compared to Nes-GFP- and endothelial cells respectively.

6) Conditional deletion of Cxcl12 in Nes-GFP+ cells led to a significant 30% reduction in CFU and a non-significant decrease in repopulating HSC.

The experiments are well performed and described. With minor exceptions the figures are clear and well presented. I have only minor suggestions and comments about the paper. The reviewer appreciates the very thoughtful and extensive discussion that attempts to resolve some apparent controversies in the field. I believe the manuscript is a significant contribution the HSC niche field. So many different transgenic strains of mice, here at least 12 gets confusing with all the acronyms and the lack of good explanations, here a table would have helped.

---

## [Author Response]

Reviewer #1:

*This manuscript reports an interesting and potentially important finding on functional heterogeneity and developmental origin of bone marrow stromal cells*.

We thank the reviewer for finding our study interesting and potentially important, and for the suggestions that have helped to improve the paper.

*However the main and potentially ground breaking finding, i.e. that Nestin+ MSC are derived from neural crest and support hematopoiesis rather than building the bone itself, is not supported by the data presented, that are of poor quality and do not allow to draw final conclusions on the developmental origin of these cells*.

We apologize if the quality of the images examined was suboptimal. Although we provided images covering wide bone marrow areas to convincingly illustrate our findings, these images also had good resolution and might have lost it during the compression and conversion to pdf. Nevertheless, we now provide insets from low magnification pictures with increased resolution. We have also shown high magnification details to facilitate viewing.

We have provided new data that further and fully support the main conclusion of our study already stated in the title that the neural crest is a source of MSCs with a specialized HSC niche function:

1) Higher detail images from triple transgenics showing neural-crest-derived cells co-labeled with Nestin-GFP, providing additional proof of a neural crest origin for a subset of Nestin-GFP+ cells (although other populations marked by Nestin-GFP+ might contain cells from other origins).

2) Functional data showing the enrichment in mesenchymal stem cell activity (CFU-F) and HSC niche features (Cxcl12 mRNA expression) within sorted bone marrow neural-crest-derived stromal cells, which are also enriched in endogenous Nestin mRNA expression.

3) Flow cytometry data showing that the majority of bone marrow cells derived from the neural crest are indeed stromal Nestin-GFP+ cells.

*Moreover, the manuscript is poorly written, difficult to follow and would greatly benefit from careful editing and, at least in some parts, re-writing*.

We apologize for this. We have re-written some parts and have revised the whole manuscript to make it more accessible to the broad readership of *eLife*. Following the suggestion by Reviewer #3, we have also included a table indicating the numerous mouse strains used, to help the reader.

*1) Authors report that a small but increasing fraction of GFP+ also express CD31 and that this fraction increases with age. Since Nestin is not known to be expressed in endothelial cells, a co-staining with also anti-nestin antibody should be performed and co-localization investigated at confocal level, since this bears important implications on the fidelity of the GFP transgene*.

Although nestin was originally identified as a neural stem cell marker in the developing central nervous system (Lendahl U et al. 1990 Cell), later studies have detected nestin expression in many other cell lineages, most pronouncedly during development (for a review, please

see Wiese et al. Nestin expression, a property of multi-lineage progenitor cells? Cell Mol Life Sci (2004) vol. 61 (19-20) pp. 2510-22). Many studies have reported nestin expression in endothelial cells, especially in proliferating vascular endothelial cells. Please see (among others):

Ono N et al. Dev Cell. 2014 May 12;29(3):330-9.

Wroblewski J et al. Differentiation. 1997 Feb;61(3):151-9.

Sugawara K et al. Lab Invest. 2002 Mar;82(3):345-51.

Mokrý J et al. Stem Cells Dev. 2004 Dec;13(6):658-64.

Kim E et al. Invest Ophthalmol Vis Sci. 2014 Jul 17;55(8):5099-108

In fact, nestin has been recently proposed as a marker for newly formed blood vessels (Matsuda et al. World J Gastroenterol (2013) 19 (1) pp. 42-8). In light of this emerging literature, we have carefully analyzed the nestin^+^ endothelial subset to avoid some potential confusion, although the population of interest was the stromal CD31^-^
*Nestin*-GFP^+^ population, which is highly enriched in mesenchymal progenitors. The other reason for using this marker to visualize blood vessels was to substantiate the perivascular or sub-endothelial localization of these mesenchymal progenitors.

Regarding the fidelity of the GFP transgene, we already showed in Figure 1 of the manuscript that *Nestin-*GFP+ cells are clearly enriched in endogenous *Nestin* mRNA expression. In the new Figure 7 we also show that bone marrow CD45^-^ CD31^-^ Ter119^-^ Tomato+ cells sorted from 1-week old *Wnt1-Cre2;R26-tomato;Nes-Gfp* triple-transgenic mice are highly enriched in endogenous *Nestin* expression, compared with bone marrow CD45^-^ CD31^-^ Ter119^-^ Tomato- cells. In the new data provided in new Figure 4 we show that the majority of neural crest cells tracked using two different drivers, *Wnt1-Cre2* and *Sox10-CreER*^*T2*^, are in fact *Nestin*-GFP+ PDGFRa+ MSC-enriched cells which contain most bone marrow mesenchymal activity (CFU-F) at early postnatal stages. All this data convincingly demonstrate endogenous nestin expression in the cell population of interest in this study.

*2) The text describing data presented in*
Figure 2
*is particularly confusing and almost impossible to interpret: “These mice and newborn Nes-gfp embryos were analyzed for osterix protein expression, which marks cells committed to the osteoblastic lineage. In contrast, cells genetically traced by the regulatory elements of Osterix gene do not only comprise osteoblastic cells, but also adventitial reticular cells, vascular smooth muscle cells, adipocytes and perineural cells (*[43]*). Unlike cells derived from lateral plate mesoderm, fetal limb bone marrow Nes-GFP+ cells did not express osterix protein (*Figure 2*) and therefore could not be considered osteoblast precursors*.*”*

Whereas osterix protein marks osteolineage cells, *Osterix* mRNA or cells marked by the regulatory elements of *Osterix* promoter do not only contain osteoblastic-committed cells, but also a variety of other cells in the bone marrow, including adventitial reticular cells, vascular smooth muscle cells, adipocytes and perineural cells. This has been clearly shown in a study cited in our manuscript (43). This is likely explained by the fact that different stem cells (including MSCs) actively transcribe differentiation factors which are not translated into proteins, which that does not imply lineage commitment but allows them to rapidly differentiate in response to stimuli. This explains some contradictions in the field. We have reworded the corresponding section to reduce confusion. Unlike mesodermal derivatives, *Nes-*GFP+ did not express highly osterix protein and they did not contribute significantly to fetal osteocytes. We therefore conclude that these cells were not osteoblast precursors (even though they might show *Osterix* mRNA expression).

*After describing the Hoxb6-CreERT mice (labelling lateral mesoderm derived cells) the Authors show no co-localization of Nes-GFP and Osterix, and then “presumed” co-localization of Hoxb6 (i.e. lateral plated) derived cells and Osterix (*Figure 2*). They conclude that because Nestin-GFP+ cells do not co-localize with Hoxb6 labelled cells they cannot be of lateral plate origin. This is a crucial point and particularly weak. First of all, at the magnification shown it is impossible to draw any conclusion. In A', very few GFP+ cells are visible and in an area where they are quite distant from Osx+ cells*.

*Hoxb6-CreERT* line would only label cells that express this transgene at the time that tamoxifen were administered. Together with limited recombination efficiency, this might explain why few GFP+ cells were observed using this mouse line and induction regimen. However, most of these cells labeled with cytoplasmic GFP also had nuclear osterix protein (please be aware of the different location of the fluorescent signal; new Figure 2 in the revised manuscript, also Figure 8 below). This is in sharp contrast with the virtual absence of osterix protein in stage-matched *Nestin-*GFP+ cells (new Figure 2).Author response image 1.Expression of osterix by bone marrow cells of the *Hoxb6*-derived lineage. High magnification detail of E18.5 *Hoxb6-CreERT2;RCE* bone marrow (BM) stained with anti-osterix (red). Embryos were induced with tamoxifen at E10.5 to trace limb mesoderm with *Hoxb6-Cre driver*. Asterisks indicate nuclear osterix-positive cells also marked with GFP. Dashed line indicated the bone contour. *Scale bar:* 50μm.

*On the other hand, in B' only a very small minority of HoxB6 derived cells express Osx (and at that magnification it is hard to be sure): in most case the colour overlap seems partial, as it may originate from two cells one above the other. Even assuming that there is co-localization in few cells, what about all the others Osx+ cells? Even though recombination is never 100% efficient, one would conclude that the very large majority of Osx expressing cells do not derive from lateral mesoderm. Maybe they originate from paraxial mesoderm, but then another Cre would be required or the origin of Osx expressing cells remains largely unidentified in this work*.

The degree of recombination using *HoxB6-CreER* driver line is only partial, which might be due both to the time window when tamoxifen was administered and to limited recombination efficiency by this Cre line. This might underestimate the real contribution of lateral mesoderm to osterix+ cells. Although it is possible, as suggested by the reviewer, that other mesodermal sources contribute to osteoprogenitor cells, this possibility would not affect the conclusions of our paper and also falls outside its current and already wide scope. We have taken the advantage of this genetic system to visualize mesodermal derivatives (in this case, lateral plate) and have demonstrated that this population, unlike *Nestin-creER*^*T2*^-traced cells, overlaps with embryonic skeletal precursors expressing osterix protein (an accepted marker of osteoprogenitor cells). We have also shown an abundant contribution of mesoderm to mature skeletal cells in long bones. This is in sharp contrast with the virtual absence of osterix protein in the *Nestin-GFP+* population and the absence of *Nestin-creER*^*T2*^-traced fetal osteoblasts or chondrocytes. Therefore the main conclusion of this part, that nestin+ cells do not significantly contribute to fetal osteochondral cells, is fully supported by our data.

*3) The overlap between Nestin-GFP and Nestin-Cre derived cells is only minor and, again, at that magnification it is impossible to draw solid conclusions. It is possible that Nestin-Cre derived cells have turned off the expression of Nestin gene, but then what have they become? Conversely, many Nestin-GFP+ do not seem to have originated from cells previously expressing Nestin. Even the example at “high” magnification (*Figure 2*) shows one cell (**) that is clearly double labelled and another example (*) where there are two neighbouring but clearly distinct cells. These data are not convincing at all of a reliable lineage tracing*.

The overlap between *Nes-*GFP+ cells and *Nes-creER*^*T2*^*-*traced cells depends on the developmental stage, tamoxifen dose and stage of administration, the time between tamoxifen administration and analysis, and the strength of the reporter mouse. We have replaced former panel 2D, based on *KFP* mice, a weak reporter line, by the Rosa26-Tomato reporter which provides a much stronger signal. *Nes-gfp;Nes-creER*^*T2*^*;Rosa26-Tomato* triple-transgenic mice show a consistent labeling and significant overlap of GFP+ and Tomato+ cells when tamoxifen is administered at perinatal stages and the analysis is performed within few days/weeks. We have provided additional images at higher magnification to better support our conclusions (new Figure 2 and Figure 9).Author response image 2.Overlap of *Nestin-GFP*+ cells with *Nestin-CreER*^*T2*^-labelled cells in the bone marrow. Representative confocal projections of skull bone marrow, showing the endogenous fluorescent signals of Tomato in red (upper right) and GFP in green (lower right); their corresponding overlay is indicated in orange/yellow. *Nestin*-*gfp*;*Nestin-CreER*^*T2*^*;R26-Tomato* triple-transgenic mice were treated with tamoxifen (4 mg, oral gavage to mother) at neonatal stage (P0) and were analyzed after 2 weeks. Asterisks indicate double-positive cells.

*4) Data on Wnt1-Cre appear more convincing, but again, a high magnification of a double-labelled cell at the confocal level would be necessary, especially for osteocytes (Cartilage staining is quite convincing even at low mag)*.

We have stained osteocytes with phalloidin in bone marrow sections from *Wnt1-Cre2;R26-tomato* mice. The pictures shown with high magnification clearly confirm the presence of phalloidin+ neural-crest-traced osteocytes in the neonatal long bones (new Figure 4). We have also included a complete section of a long bone from *Nes-Gfp;Wnt1-Cre2;R26-tomato* triple-transgenic mice that does not only illustrate neural-crest-derived chondrocytes in the outermost layer of both diaphyses, but also abundant neural-crest-derived cells in the endosteal bone marrow region (new Figure 4).

*However, FACS analysis shown in*
Figure 4
*has problems. The Wnt1-Cre/Tomato clearly shows a dim (continuous with background) and a very bright population, both gated together to show that 62% of these are Nes-GFP+. I would have liked to see whether the Nes-GFP+ cells are equally represented in the bright and the dim populations. This is critical because the results are highly dependent on the specific gating*.

We have provided the negative control to substantiate the gating strategy (new Figure 4). We have also reanalyzed the data and have performed additional experiments in *Wnt1-Cre2;R26-Tomato;Nes-Gfp* triple-transgenic mice analyzed at neonatal stage. As the reviewer states, two stromal populations can be distinguished based on Tomato fluorescence intensity. Although both Tomato+ stromal populations contain *Nes-*GFP+ PDGFRα+ MSC-enriched cells, these cells are progressively enriched in the Tomato^dim^ population with age (new Figure 4—figure supplement 2).

*Most importantly, the IF shown in*
Figure 4
*is highly unconvincing and does not support the claim that Nes-GFP+ are derived from Wnt1 expressing cells. The very few Tomato+ cells do not really co-localize with GFP+ cells and the examples shown in D look like at most closely located cells*.

We admitted in our manuscript the heterogeneity of the *Nes*-GFP+ population, which does not only contain neural-crest-derived cells but also endothelial cells. In the revised manuscript we have provided more clear images showing *Nes-*GFP+ Tomato+ cells (new Figure 4 and Figure 10).Author response image 3.High magnification confocal image from 1-week old *Wnt1-Cre2;Tomato;Nes-Gfp* BM section, showing co-localization of NC-derived Tomato+ cell with Nes-GFP (arrowhead).

We have performed additional experiments that fully support our main message that neural-crest-derived nestin+ MSCs have a specialized function in forming the HSC niche in the bone marrow. We have analyzed *Cxcl12* and *Nestin* mRNA expression in Tomato^+/-^ GFP^+/-^ bone marrow stromal cells sorted from P7 *Wnt1-Cre2;R26-Tomato;Nes-Gfp* triple-transgenic mice. Among neural-crest-traced cells, *Nes*-GFP^+^ BMSCs were particularly enriched in the expression of *Cxcl12* and endogenous *Nestin* (Figure 7). We have measured MSC activity (CFU-F) in Tomato^+/-^ GFP^+/-^ bone marrow stromal cells sorted from P7 *Wnt1-Cre2;R26-Tomato* double-transgenic and P7 *Wnt1-Cre2;R26-Tomato;Nes-Gfp* triple-transgenic mice. When plated at equal cell density, colonies were only detected in the neural-crest-derived cells (new Figure 4 and Figure 4—figure supplement 1). These results fully support our original contention.

*Finally, the FACS analysis for Sox10-Cre also shows that gating has included both bright cells, clearly separated from the negative peak and the shoulder of this. I would like to see whether the bright cells only are GFP+*.

The reviewer is correct that different levels of Tomato intensity were also detected in *Sox10-creERT2* mice, but both Tomato^bright^ and Tomato^dim^ bone marrow stromal populations stromal contained *Nes-*GFP+ PDGFRα+ MSC-enriched cells. We have reanalyzed our data gating on the Tomato^bright^ population, as requested. The Tomato^bright^ population contains a high percentage of *Nes-*GFP+ cells, also including PDGFRα+ cells (new Figure 4—figure supplement 2).

*5) The final sections of the Results, though also suffering of the poor quality of the images, are more convincing but less original. Furthermore DTR experiments are tricky, first because killing Nestin expressing cells may have also consequences in other tissue and possibly systemic effects. In addition, and this criticism applies to all of these studies, developmental biologists seem to be unaware of the “bystander effect”, well studied by gene therapists, where killing a specific cell, often causes the death of the nearest cells, thus mudding the results*.

We hope that the revised images show the quality level expected by the reviewer and us. We agree that selective cell depletion experiments might have some bystander effects; for this reason we already showed in our manuscript the normal bone marrow histology and blood vessel architecture of experimental *Nes-creERT2;iDTA* mice (Figure 7—figure supplement 1) as a proof of the absence of noticeable bystander effects. To further substantiate this, we have included in the response to the reviewer bone marrow sections of *Nestin-CreER*^*T2*^;*iDTR* and control *iDTR* mice injected with tamoxifen and diphtheria toxin to demonstrate that cell deletion using this inducible Cre line is not associated with noticeable abnormalities in bone marrow histology or vascular leakage of FITC-dextran (green, please see below) previously intravenously injected in the mice (Figure 11).Author response image 4.Intravital microscopy of mice previously injected with FITC-dextran (green) to demonstrate the absence of vascular leakage (upper panels). H & E staining of bone marrow sections showing a normal histology of the bone marrow (lower panels).

Reviewer #2:

*The manuscript by Isern et al describes the results of experiments examining the developmental origins (fetal, neonatal and adult) of the mesenchymal cells that are osteogenic and that support the hematopoietic system in the bone marrow. The experiments mainly utilize the nestin-gfp reporter mouse model in addition to other mouse models in which induced recombination mediated reporter expression defines specific neural crest derived populations. The manuscript is filled with information but it is sometimes difficult to understand how it is related to the main question. For example, it is not explained why CD31 immunostaining is performed on nestin gfp tissues; what cell types does it mark? Convention is the endothelial and hematopoietic cells express CD31, not MSC. The Discussion provides some information about levels of CD31 expression but this comes rather late in the manuscript*.

We are currently lacking robust and specific MSC markers for immunofluorescence. We have used CD31 marker for two reasons: 1) To mark blood vessels, since *Nes-*GFP+ cells are closely associated with them. 2) To distinguish the small fraction of perinatal bone marrow *Nes-*GFP+ CD31+ cells from the majority of *Nes-*GFP+ CD31- cells, which contain the population of interest in our study. Please see also Response to Major Comment #1 by Reviewer 1.

*In the Results section that describes BM nestin+ cells do not contribute to fetal endochondrogenesis, the RCE reporter is used, what is this? No definition is provided*.

The RCE acronym stands for *Rosa26 CAG EGFP*, an *EGFP* reporter allele which is expressed upon *Cre*-mediated recombination from the hybrid CAG promoter in the *Rosa26* locus (Sousa et al. Characterization of Nkx6-2-Derived Neocortical Interneuron Lineages. Cerebral Cortex (2009) vol. 19 (Supplement 1) pp. i1-i10). We have defined it as “…a sensitive reporter that drives stronger GFP expression than other reporter lines (71)…” .

*Also, later in this paragraph, the sentence beginning with “In contrast, cells…” seems misplaced and confusing, making this text and results hard to follow*.

We have moved this to the Discussion and have re-written this paragraph. Please see response to the similar Major Comment #2 by Reviewer 1.

*Similarly, it is not understood why a Wnt Cre line was used. What is the rationale? I seem to miss the connection. It would be helpful to reorganize and perhaps move some information from the main text to the Discussion or leave it out if it is not directly related to the question. It would be helpful to the reader if the manuscript could be more direct in describing the experimental rationale*.

We apologize for not being sufficiently clear. We have rewritten the manuscript to better explain the rationale of each single experiment and have made an effort to better link contiguous sections.

*Altogether, the authors have done a large amount of work to show that there are different component populations of mesenchymal cells that contribute to the development of the long bones, bone marrow and the supportive hematopoietic microenvironment. This is an important set of results that supports a new conceptual model for the development of the hematopoietic supportive microenvironment*.

We thank the reviewer for considering that our results importantly support a new conceptual model for the establishment of stem cell niches.

Reviewer #3:

*This manuscript identifies the ontological source of an important bone marrow (BM) niche cells previously identified by the senior author. These cells are the Nestin-GFP+ (Nes-GFP+) cells in the adult BM that are a source of MSC and niche support. The Mendez-Ferrer group has extended this work to the developing fetal and early postnatal bone and observed exciting and enlightening results. They demonstrate that trunk neural crest cells are the developmental source for Nes-GFP+ MSC in the BM, in addition to their known contributions to peripheral sympathetic neurons and Schwann cells. Interestingly, all three of these cells types are important BM niche components. As such they share an ontological relationship that could theoretically extend to other niche components elsewhere in the body. Nes-GFP+ cells were characterized in the developing bone (E17.5), at P0 and at P7 with a variety of lineage tracing and Cre deletion transgenics in addition to cell surface IHC and flow cytometry. From a wide array of models they show*:

*1) Nes-GFP+ cells do not exhibit osteochondral progenitor activity. Nes-GFP- cells that derive from lateral plate mesoderm possess this activity in fetal bone*.

*2) Neural crest contribution to postnatal BM-MSCs was definitively demonstrated by genetic fate mapping with Wnt1-Cre2 and Sox10-CreERT2*.

*3) Cell surface and mRNA-seq studies of PDGFRa+/- and Nes-GFP+/- cells revealed that there are two Nes-GFP+ neural crest derived cells in postnatal BM. Schwann cell precursors are in the PDGFRa- fraction while PDGFRa+ cells contain the MSCs*.

*4) Neural crest cells migrate along developing nerves to the fetal bone before commitment to the Schwann cell lineage and give rise to niche forming MSCs*.

*5) Cxcl12 expression was shown to be 80 and 20-fold higher in Nes-GFP+ cells compared to Nes-GFP- and endothelial cells respectively*.

*6) Conditional deletion of Cxcl12 in Nes-GFP+ cells led to a significant 30% reduction in CFU and a non-significant decrease in repopulating HSC*.

*The experiments are well performed and described. With minor exceptions the figures are clear and well presented. I have only minor suggestions and comments about the paper. The reviewer appreciates the very thoughtful and extensive discussion that attempts to resolve some apparent controversies in the field. I believe the manuscript is a significant contribution the HSC niche field. So many different transgenic strains of mice, here at least 12 gets confusing with all the acronyms and the lack of good explanations, here a table would have helped*.

We thank very much the reviewer for appreciating the relevance of our study and for the very positive criticism. We have included a table containing a detailed summary of all the mouse strains used in supplementary procedures (Supplementary file 3).